



# Regional Interaction Frameworks to Support Multi-Hazard Approaches to Disaster Risk Reduction (With an Application to Guatemala)

Joel C. Gill[1], Bruce D. Malamud[2], Edy Manolo Barillas[3], Alex Guerra Noriega[4]

5   [1]Global Geoscience, British Geological Survey, Keyworth, NG12 5GG, UK. Corresponding Author: joell@bgs.ac.uk
[2]Department of Geography, King's College London, London, WC2B 4BG, UK
[3]UN Office for the Coordination of Humanitarian Affairs, Guatemala City, Guatemala
[4]Instituto Privado de Investigación sobre Cambio Climático, Guatemala City, Guatemala

10   *Correspondence to*: Joel C. Gill (joell@bgs.ac.uk)

**Abstract.** Here we present an interdisciplinary approach to developing comprehensive, systematic and evidenced regional interaction frameworks to support multi-hazard approaches to disaster risk reduction. We apply this approach in Guatemala, developing regional interaction frameworks for national and sub-national (Southern Highlands) spatial extents. The regional interaction frameworks are constructed and populated using five evidence types: (i) publications and reports (internationally accessible) (93 peer-review and 76 grey literature sources); (ii) publications and reports (locally accessible civil protection bulletins) (267 bulletins from 11 June 2010 to 15 October 2010); (iii) field observations; (iv) stakeholder interviews (19 semi-structured interviews) (v) stakeholder workshop results (16 participants). These five evidence types were synthesised to determine an appropriate natural hazards classification scheme for Guatemala, with 6 natural hazard groups, 19 hazard types, and 37 hazard sub-types. For a national spatial extent in Guatemala, we proceed to construct and populate a regional interaction framework (matrix form), identifying 50 possible interactions between 19 hazard types. For a sub-national spatial extent (Southern Highlands of Guatemala), we construct and populate a regional interaction framework (matrix form), identifying 114 possible interactions between 33 hazard sub-types relevant in the Southern Highlands. We also use this evidence to explore networks of multi-hazard interactions and anthropogenic processes that can trigger natural hazards. We present this information through accessible visualisations to improve understanding of multi-hazard interactions in Guatemala. We believe that our regional interaction frameworks approach to multi-hazards is scalable, working at global to local scales with differing resolutions of information. Our approach can be replicated in other geographical settings, with regional interaction frameworks helping to enhance cross-institutional dialogue on hazard interactions, and their likelihood and potential impacts.



# 1 Introduction

## 1.1 Regional Interaction Frameworks

This paper develops comprehensive, systematic and evidenced *regional interaction frameworks* to support multi-hazard approaches to disaster risk reduction (DRR). We apply our interdisciplinary approach to two regional spatial extents in Guatemala (national and sub-national), developing what we believe to be the first national scale comprehensive characterisation of potential hazard interactions in the published literature, relevant to a wide range of actors involved in DRR. This approach is scalable and replicable in diverse contexts, as we discuss throughout this paper.

Here we define regional interaction frameworks to be visualisations that support the identification and characterisation of relevant hazard interactions in a defined region (from $10^2$ to $10^6$ km$^2$). Interaction frameworks are a component of a 'multi-hazard' approach, defined by the UN Office for Disaster Risk Reduction (UNISDR) as *"the selection of multiple major hazards that the country faces, and the specific contexts where hazardous events may occur simultaneously, cascadingly or cumulatively over time, and taking into account the potential interrelated effects"* (UNISDR, 2017).

Here, and throughout this paper, we use 'hazard interactions' to describe these cascades and interrelated effects. Examples include tropical storms triggering floods and/or landslides; volcanic eruptions triggering wildfires that subsequently increase the probability of debris flows occurring; and earthquakes triggering regional subsidence. Many more examples, and extensive case studies, feature in the literature (e.g., Tarvainen *et al.,* 2009; Kappes *et al.,* 2010; Gill and Malamud, 2014; Duncan *et al.,* 2016). The UNISDR Sendai Framework for Disaster Risk Reduction (SFDRR) emphasises the need for multi-hazard approaches (UNISDR, 2015). Stakeholders involved in implementing the SFDRR (e.g., civil protection agencies, hazard-monitoring scientists, urban planners, development practitioners) will therefore all benefit from frameworks that systematically identify and characterise hazard interactions in regional contexts.

There are limited examples of *regional interaction frameworks* in the literature. **Table 1** outlines and characterises seven examples of frameworks for specific named regions or geographical features that include natural hazards and a deliberate attempt to characterise possible hazard interactions. While there is significant variation in the approaches used to construct and populate these frameworks, they helpfully demonstrate the scalability of *regional interaction frameworks*. Building on these examples, we present and apply a methodology in this paper to develop and enhance *regional interaction frameworks*. We integrate six themes, identified by Gill (2016), of *spatial scale*, *temporal scale*, *likelihood-magnitude relationships*, *selection* and *classification of natural hazards*, *identifying relevant hazard interactions*, and *visualisation style and user communities* to ensure frameworks are comprehensive, systematic and evidenced. While the frameworks developed in this paper specifically support Guatemalan stakeholders, we suggest that our approach supports implementation of the SFDRR in other settings. Our approach is replicable and can help to improve characterisation of hazard interactions.



## 1.2 Application to Guatemala

We trialled our approach in Guatemala due to (i) the hazardousness of the region, and (ii) logistical feasibility (contacts, language, accessibility). A broad range of natural hazards and anthropogenic processes in Guatemala make it an appropriate country to examine hazard interactions. Guatemala's dynamic geological history and geographical setting give rise to many potential hazards. These include geological (e.g., earthquakes, volcanic activity, landslides and surface collapses) and hydrometeorological hazards (e.g., tropical cyclones, thunderstorms, hailstorms, tornados, coastal storm surges, floods, drought, heatwaves and cold spells), defined by UNISDR (2017). Guatemala ranks high in descriptions of countries exposed to multiple hazards and risks (e.g., Welle *et al.,* 2013; Kreft *et al.,* 2015; Bündnis Entwicklung Hilft/United Nations University, 2017).

Two principal government organisations exist in Guatemala, tasked with supporting DRR. These are CONRED (Coordinadora Nacional para la Reducción de Desastres/National Coordinator for Disaster Reduction) and INSIVUMEH (Instituto Nacional de Sismología, Vulcanología, Meteorología e Hidrología/National Institute for Seismology, Volcanology, Meteorology and Hydrology). **Table 2** gives an overview of these organisations. Additional organisations engaged in research and practitioner work relating to natural hazards and DRR include universities (e.g., Universidad de San Carlos de Guatemala), private sector consultancies and research institutes (e.g., Private Institute for Climate Change Research), civil society organisations (e.g., Oxfam), and regional and international intergovernmental organisations (e.g., CEPREDENAC, UN OCHA).

## 1.3 Structure of Paper

In **Section 2** we characterise five diverse evidence types. We integrate and use this evidence in **Section 3** to characterise hazard interactions and networks of interactions (cascades), constructing two regional interaction frameworks for Guatemala (national and sub-national spatial extents). In **Section 4** we discuss our findings in the context of regional interaction frameworks and multi-hazard assessments, with conclusions in **Section 5**.

## 2 Evidence Used to Inform the Regional Framework

### 2.1 Evidence Types and Integration

Developing comprehensive and evidenced *regional interaction frameworks* requires diverse evidence to improve the systematic identification of relevant hazards and interactions. In **Table 3**, we group possible evidence types into (1) publications and other reports, (2) social and other media, (3) field evidence, (4) stakeholder engagement, and (5) miscellaneous. Some overlap exists between these categories, and not all the examples given are relevant in any given location. Of the evidence types in **Table 3**, we used the following:





- *Publications and reports (internationally accessible)* (**Section 2.2**). A comprehensive synthesis of literature describing natural hazards in Guatemala and their interactions. This included peer-review material, technical reports, databases and media reports.

- *Publications and reports (locally accessible)* (**Section 2.3**). Analysis of government issued, Spanish-language, civil protection information bulletins.

- *Field observations* (**Section 2.4**). Reconnaissance trips, giving an overview of the Guatemala's hazard-forming environment (defined in Liu *et al.,* 2016).

- *Stakeholder engagement* (**Sections 2.5** and **2.6**). 19 semi-structured interviews and a 3-hour workshop with hazard and civil protection professionals in Guatemala.

Other evidence types (e.g., historical records, community knowledge) are included in peer-review and grey literature publications we examined. For each evidence type considered, we do not use all possible examples, methods and sources; rather we use examples of key case studies from regions of interest. In **Table 4**, we summarise how we integrate evidence types to develop our regional interaction frameworks.

## 2.2 Publications and Reports (Internationally Accessible)

Internationally accessible publications and reports includes both peer-review and grey literature, including journal articles, edited volumes, Masters and PhD theses, textbooks, technical reports, databases, and NGO disaster situation reports. These all report hazard events in specific geographic regions, providing evidence of interactions. For example, Rose *et al.* (2004) present a set of papers on natural hazards in El Salvador (edited volume), and ReliefWeb (2018) present a situation report on the impact of Tropical Storm Nate in Central America (disaster situation report). We identified multiple publication and report types with information about Guatemala. We prioritised literature giving a broad overview of natural hazards, synthesising multiple texts, or characterising hazard interactions. It is beyond the scope of this study to examine publications on every aspect of hazards in Guatemala, or to review all publications on any one aspect of a hazard.

We primarily accessed literature using large web-databases (Google Scholar, Web of Science) for peer-reviewed articles and general online searches for other grey literature (e.g., media reports). We used Boolean search methods, including both 'Guatemala' and keywords associated with a preliminary list of 21 natural hazards (from Gill and Malamud, 2014). For example, 'earthquake', 'aftershock', 'seismic', 'tremor', and 'liquefaction' were searched for alongside 'Guatemala' and 'Central America' to identify relevant material. We evaluated results to determine their relevance and identify other keywords. We also identified specialist books, such as an edited volume on the geology of Central America (Bundschuh and Alvarado, 2007).



We examined literature in a systematic manner, collating references, maps and figures for 17 (of the 21) natural hazards: earthquake, tsunami, volcanic eruption, landslide, flood, drought, regional subsidence, ground collapse, soil (local) subsidence, ground heave, storm, tornado, hailstorm, lightning, extreme temperature (heat), extreme temperature (cold), and wildfire. Snow avalanches and snowstorms have limited spatial relevance in Guatemala, and geomagnetic storms and

impact events have little country-specific (vs generically relevant) information. For each hazard considered, we cross-referenced diverse literature to characterise it at a level of detail appropriate to this study, including information on spatial and temporal distribution, triggering relationships, and impacts. We identified and used as evidence 169 sources, with 93 (55%) of these being peer-review, and 76 (45%) of these being grey literature. We use this evidence in **Section 3** to help develop a regionally appropriate hazard classification and synthesise relevant interactions.

**2.3 Publications and Reports (Locally Accessible)**

Another evidence type to inform the development of regional interaction frameworks is locally accessible reports, such as government or NGO bulletins, newspapers, and emergency call out records. Civil protection information bulletins and newspapers can both give a focused overview of natural hazard occurrences (e.g., Guzzetti *et al.* 1994; Trimble, 2008; Raška *et al.* 2014; Taylor *et al.,* 2015), providing information on hazard interactions or noting triggering relationships.

In Guatemala, we use Spanish-language civil protection information bulletins from the *Coordinadora Nacional para la Reducción de Desastres* (CONRED, National Coordinator for Disaster Reduction). Bulletins are issued when there is a threat to lives, livelihoods, and infrastructure, and include information on hazards, their spatial and temporal extent, and their impacts, including hazard triggering. Natural hazards occurring in remote regions or having a very low impact (e.g., very small landslides) are unlikely to be included in bulletins, and therefore bulletins do not provide a complete record of

events. CONRED may issue multiple bulletins per day, depending on the evolution of, for example, a weather system or a disaster event. Bulletins are distributed to a mailing list of personnel, with some on their website (CONRED, 2018b) and ReliefWeb (2016). At the time of writing, CONRED bulletins are not systematically archived online. We therefore classify these bulletins as locally accessible.

CONRED made available to the authors (electronic format) 267 accessible information bulletins published over a 127-

25   day period between the 11 June 2010 and 15 October 2010. Based on their numbering, we believe CONRED published 413 bulletins during this 127-day period. Additional information that characterises these bulletins is included in **Supplementary Material (Table S1)**. We searched the 267 accessible bulletins for keywords, placing these into context by looking at the surrounding sentences. Taylor *et al.* (2015) used this approach to enrich the UK national landslide database by examining newspaper archives.

We selected and used the following six keyword verbs connecting two hazard types and suggesting an interaction between them (with an abbreviated Spanish verb base in parentheses): *to trigger* (*desenca*), *to provoke (provoc)*, *to generate*



*(genera)*, *to cause (caus)*, *to produce (produ)*, and *to catalyse* (*catal*). We performed a keyword Boolean search in Spanish using the abbreviated form of the verb base to ensure the return of multiple derivatives of the verb. To check if there were other verbs of interest, we then searched for the following hazard keywords in Spanish form (both singular and plural): *seismic, earthquake, volcano, eruption, landslide, flood, collapse, sinkhole, hurricane, storm, tsunami, drought, tornado,*

*wind, rain.* We also searched for references to three active volcanoes (*Pacaya, Santiaguito,* and *Fuego*) in Guatemala. From these hazard keywords and three volcanoes, we looked for any further interaction verbs that might be included near these words and identified no additional keyword verbs using these. In **Table 5**, we show a summary of keywords used and a total of 143 results over 95 bulletins, prior to processing based on relevance. These results included some bulletins with more than one result.

By examining the context, we determined that 39 of the 143 results (from 36 different bulletins, on 28 unique days) described unique events where interactions occurred between natural hazards. These results are presented in the **Supplementary Material** (**Table S2**). In **Section 3.4**, we outline and characterise examples of networks of interactions extracted from these bulletins. The results in this section, although based on an incomplete dataset, demonstrate examples of the types of interactions that could occur. Further research could use a larger sample of bulletins to better characterise

interactions in Guatemala, or an event database such as EM-DAT (CRED, 2018). This would be necessary if the frequency of different types of events was a consideration, with a four-month period being too short to analyse this.

**2.4 Field Observations**

Field observations can also help to understand the relevance and dynamics of hazards and hazard interactions. For example, Havenith *et al.* (2003) describe field evidence of earthquake-triggered landslides in the Northern Tien Shan

Mountains of Kyrgyzstan. Approaches include reconnaissance visits to improve contextual understanding of the region, detailed geological, geomorphological or hazard mapping, and the application of technologies such as rain gauges, drones, and thermal imaging infrared cameras.

In Guatemala, from January to March in 2014 (9 weeks total), we visited regions in the Southern Highlands of Guatemala affected by multiple natural hazards and anthropogenic activity. We aimed to familiarise ourselves with the features of

key locations and hazards in Guatemala, but did not gather primary field data (e.g., community interviews). We enhanced our:

    i.   *Understanding of Guatemala's hazard environment*, observing the spatial and temporal scales at which hazards and anthropogenic processes act.



ii.   *Interviews with expert participants* (**Section 2.5**), collecting richer data because of a clearer understanding of examples used by participants to evidence natural hazard interactions in Guatemala. Visits helped identify local place names and descriptors for sites affected by hazards.

We conducted field visits alongside INSIVUMEH, with support from the University of Bristol. This helped to develop

constructive relationships, establishing the mutual trust and respect required for subsequent data-rich interviews (Kitchin and Tate, 2000). We completed one field visit with CONRED. In **Table 6**, we use personal observations and information from both peer-review and grey literature to describe select locations. We include information on interactions and/or networks of interactions. We return to these examples when illustrating networks of hazard interactions in Guatemala (**Section 3.4**).

**2.5 Stakeholder Engagement: Interviews**

Interviews provide additional evidence to construct and populate regional interaction frameworks. Participants often come from diverse backgrounds, with differing understanding of natural hazards and geographic regions. Participants with relevant evidence can include both 'experts' (e.g., hazard and disaster professionals) and local people who might be impacted by hazards (e.g., farmers, local government, communities). Selecting participants based on their experience and

relevance to a research question (purposeful sampling), can result in data-rich interviews (MacDougall and Fudge, 2001; Longhurst, 2003; Suri, 2011; Palinkas *et al.,* 2015). Semi-structured interviews provide one means by which to have this dialogue, focused around questions on hazards and hazard interactions. This style gives enhanced freedom to explore areas of interest and pursue emerging lines of enquiry (Qu and Dumay, 2011).

Prior to travelling to Guatemala in 2014, we obtained ethics approval from King's College London for research with

human participants. At the start of each interview we explained the purpose of our work and sought informed, prior consent to use data generated. All participants gave permission for us to use their data and identify their institution unless this would identify the individual. We interviewed 21 hazard and civil protection professionals in Guatemala, during 19 interviews. **Supplementary Material (Table S3)** characterises the interview participants. Participants came from academia, the private sector, INSIVUMEH and CONRED. We selected interview participants from diverse professional

backgrounds in terms of hazard speciality (e.g., earthquakes, landslides, floods) and engagement in the disaster cycle (e.g., early warning, mitigation, recovery). We identified contacts before travelling to Guatemala through their online profiles and professional engagement in other projects, and through introductions once in Guatemala.

We ensured that participants were comfortable to reduce possible power relations between the interviewer and participant (Kitchin and Tate, 2000; DiCicco-Bloom and Crabtree, 2006; Qu and Dumay, 2011). Interviews ranged from 30−120

minutes, following a semi-structured approach (Longhurst, 2003; Qu and Dumay, 2011). Interviews included opportunities for participants to talk about (i) their background and training, (ii) their consideration and use of information





on hazard interactions, (iii) examples of existing networks of hazard interactions, and (iv) hazard interaction visualisations. All interviews aimed to cover these key themes, however there were differences in the order that they were introduced, and the specific questions asked.

**Supplementary Material** (**Table S4**) presents key statements relating to natural hazards, interactions, and anthropogenic processes, extracted from these 19 semi-structured interviews. Multiple participants highlighted specific interaction examples. These include ones noted in internationally accessible publications (e.g., lahars from Santiaguito triggering flooding, Harris *et al.,* 2006), and interactions not described in other evidence types (e.g., Pacific coastal flooding due to simultaneous high tides and river sedimentation). We use participants' comments as evidence when constructing regional interaction frameworks for Guatemala (**Section 3**), helping to develop a natural hazards classification and identify relevant hazard interactions.

### 2.6 Stakeholder Engagement: Workshop

Another form of stakeholder engagement are workshops designed to generate data through activities and focused discussion. We organised a 3-hour workshop in Guatemala involving 16 civil protection professionals at CONRED. Participants included senior and junior staff working in diverse departments. **Supplementary Material** (**Table S3**) characterises participants, with all giving permission for us to use their data in an anonymised form. Workshop limitations are discussed in **Section 4.1**.

During our workshop, participants independently completed two tasks.

Task 1.  *Network Linkage Diagram for 21 Hazards (16 participants)*. Participants used this to record triggering relationships that they believed to be relevant to Guatemala. We did not expect any participant to map out all relevant interactions.

Task 2.  *7 × 11 Hazard Interaction Matrix (15 participants)*. Participants completed a blank hazard interaction matrix, with seven primary hazards on the vertical axis and eleven secondary hazards on the horizontal axis. Results are outlined in **Section 2.6.2**.

We therefore collected two sets of visual records that document participants' perceptions of relevant hazard interactions in Guatemala. We include an example of each diagram in **Figure 1**, with all completed diagrams included in the **Supplementary Material** (**Figures S1** and **S2**). Completed network linkage and interaction matrix diagrams vary in the number and range of interactions proposed to be relevant in Guatemala. The number of interactions proposed by any one participant using the hazard linkage diagram, for example, ranged from 8 to 35, with a mean of 18 and a median (50[th] percentile) of 15.





Using all 16 completed network linkage diagrams, we can represent the combined knowledge of the workshop participants, and use this as evidence when constructing regional interaction frameworks for Guatemala. In **Figure 2**, we overlay evidence from 16 completed network linkage diagrams on a blank interaction framework, showing the number of participants (out of 16) proposing each triggering relationship. Of a total possible 441 (21×21) interactions, there are

86 different interactions proposed in **Figure 2** as being relevant in Guatemala (by 1–16 participants), equivalent to 20% of the 441 possible interactions. Consequently, 355 interactions (80% of the 441 possible interactions) were determined by all 16 participants as not relevant in Guatemala. Some of the proposed interactions may not be relevant (false positives), and others not proposed by participants may be relevant (false negatives) in Guatemala. We present more detailed statistics resulting from this workshop, and analysis of the hazard interaction matrices, in the **Supplementary Material**.

These results highlight different opinions on which hazard interactions are relevant in Guatemala. There is strong consensus on the occurrence of some interactions, but weak consensus on others. We use this data in **Section 3** as additional evidence of possible hazard interactions in Guatemala. The workshop results demonstrate the need for communication across hazard disciplines, and the value of comprehensive, systematic and evidenced frameworks to enhance understanding of relevant interactions.

**2.7 Summary of Evidence Types Used to Inform Our Frameworks**

**Sections 2.2** to **2.6** describe five evidence types (letters **A**–**E** below, and used throughout the remainder of this paper) that can help construct and populate a regional interaction framework for Guatemala:

A.   Publications and reports (internationally accessible) (93 peer-review, 76 grey literature) (**Section 2.2**).
B.   Publications and reports (locally accessible civil protection bulletins) (267 bulletins from 11 June 2010 to 15 October
2010) (**Section 2.3**).
C.   Field observations (four sites discussed in the text) (**Section 2.4**).
D.   Interviews (19 interviews, conducted from 28 February to 14 March 2014) (**Section 2.5**).
E.   Workshops (16 participants, 06 March 2014) (**Section 2.6**).

Other evidence (e.g., social media, instrumental records, and others noted in **Table 3**) may be pertinent in other
geographical locations. The use of multiple evidence types (vs. a reliance on one evidence type) facilitates a more comprehensive characterisation of hazards and hazard interactions.

**3 Regional Interaction Frameworks (Visualisations)**

We now proceed to develop our comprehensive, systematic and evidenced *regional interaction framework* for Guatemala. In **Section 3.1**, we discuss the construction and population of regional interaction frameworks. In **Section 3.2**, we present



a revised hazards classification scheme for Guatemala. In **Section 3.3**, we use this scheme and additional evidence to populate two regional interaction frameworks, a 21×21 hazard interaction matrix completed for a national spatial extent (Guatemala), and a 33×33 hazard interaction matrix completed for a sub-national spatial extent (Southern Highlands of Guatemala). In **Section 3.4**, we use these frameworks and evidence from **Section 2** to illustrate two networks of hazard interactions (cascades). In **Section 3.5**, we consider anthropogenic processes triggering hazards and catalysing interactions in Guatemala.

### 3.1 Guiding the Construction and Population of Regional Interaction Frameworks

The construction of comprehensive and systematic regional interaction frameworks requires three components for a region of interest, each bringing together diverse strands of evidence, and unifying them within a formal structure, supported by expert knowledge (Neri *et al.,* 2008):

i. **Information on relevant single hazards and appropriate ways to classify these**. Here we identify single hazards through diverse evidence types, including literature, field observations, semi-structured interviews, and workshops.

ii. **Information on relevant hazard interactions to populate the interaction framework** (i.e., identifying how single hazards interact with each other). We identify interactions using the same diverse evidence types, supplemented by literature on global and regional interaction frameworks.

iii. **An appropriate visualisation framework to represent hazard interactions.** We adapt existing visualisation frameworks (Gill and Malamud, 2016), and ensure these are appropriate to Guatemala.

Gill (2016) identified six themes for consideration when developing regional interaction frameworks: (i) spatial scale, (ii) temporal scale, (iii) likelihood-magnitude relationships, (iv) selection and classification of natural hazards, (v) identification of hazard interactions, and (vi) user requirements and visual style. We revisit these themes in **Table 7** to guide the generation of regional interaction frameworks for Guatemala. We integrate perspectives from hazard and civil protection professionals in Guatemala (from semi-structured interviews and the workshop, see **Sections 2.5** and **2.6)**. Professional organisations have an understanding of local culture, language and knowledge, and have the mandate to adapt interaction frameworks into suitable forms for other stakeholders (e.g., policy makers and communities).

### 3.2 Relevant Natural Hazards and Hazards Classification

Gill and Malamud (2014) propose a broad classification of 21 natural hazards, in six hazard groups (geophysical, hydrological, shallow Earth, atmospheric, biophysical, space). This, or an alternative, comprehensive classification can be adapted to develop a regionally specific classification, using available evidence. We use this approach to propose a





detailed, location-specific classification of natural hazard types in Guatemala, building on evidence in **Section 2**. We begin by identifying which of the 21 natural hazards listed in Gill and Malamud (2014) are relevant in Guatemala, and sub-divide selected hazards where evidence supports an expanded classification. We present our evidenced classification scheme in **Table 8**, including six hazard groups, 19 hazard types, and 37 hazard sub-types. We also include an indication

of the evidence supporting this classification, using identifying letters **A–E** introduced in **Section 2.7**, and specific referenced publications and reports where appropriate. The 37 detailed natural hazard sub-types in **Table 8** helps to improve the detail by which we can characterise hazard interactions.

Our classification is one way of grouping relevant natural hazards, with alternative classifications possible. Other natural hazard types may exist in Guatemala that have been missed from our classification, including those occurring less

frequently or having a lesser impact than those we consider. We reduce the likelihood of missing key hazards by reviewing multiple evidence types to ensure a comprehensive and evidenced classification. We include 26 to 32 more hazard sub-types than existing regional interaction frameworks (e.g., Tarvainen *et al.,* 2006; Kappes *et al.,* 2010; Liu *et al.,* 2016). In addition to the 37 natural hazard sub-types in **Table 8**, we could also consider how a changing climate influences natural hazards (see McGuire and Maslin, 2012, for a full discussion), or other groups of processes, such as biological hazards

(e.g., epidemics), technological hazards (e.g., structural collapse), and anthropogenic processes (e.g., vegetation removal). The latter are discussed in **Section 3.5**.

### 3.3 Guatemala Interaction Frameworks

Building upon the reflections in **Section 3.1**, and using the hazard classification in **Section 3.2** and evidence in **Section 2**, we now construct and populate interaction frameworks for two different spatial extents in Guatemala.

1. **National** spatial extent (**Section 3.3.1**). We produce a 21×21 interaction framework (matrix form), with 19 relevant hazards. We initially constrain interactions for a national spatial extent using the coarser hazard classification (21 hazard types).

2. **Sub-national** (Southern Highlands of Guatemala) spatial extent (**Section 3.3.2**). We produce an interaction framework (matrix form) using our classification of 37 hazard sub-types, giving a maximum of 37 primary and

25 37 secondary hazards. We use information from **Section 2** to: (a) explain and justify the selection of the Southern Highlands of Guatemala; (b) determine which of the 37 hazard sub-types are relevant in this spatial extent; and (c) adapt the 21×21 interaction framework to incorporate these hazard sub-types and populate this framework with relevant hazard interactions.

Both interaction frameworks use a matrix visualisation approach.



### 3.3.1 Guatemala National 21×21 Interaction Framework (Matrix Form)

To develop an interaction framework for the national spatial extent of Guatemala, we start with an existing 21×21 matrix (Gill and Malamud, 2014). From **Table 8** we identify that 19 of the 21 natural hazards in this matrix are relevant to Guatemala. Using the evidence in **Section 2**, we systematically examine each matrix cell to consider whether an interaction is possible in Guatemala. We present our completed national-scale, regional interaction framework in **Figure 3**, with 21 primary natural hazards on the vertical axis (of which 19 are relevant), and the same 21 secondary (of which 19 are relevant) natural hazards on the horizontal axis. 50 (11%) of 441 cells are shaded, indicating 50 possible interactions. These include:

i.   *Triggering Only*. 15 (30%) of the 50 interactions.

ii.  *Increased Probability Only.* 5 (10%) of the 50 interactions.

iii. *Triggering and Increased Probability.* 30 (60%) of the 50 interactions.

The evidence types (**A–E**) supporting these 50 hazard interactions is outlined in **Supplementary Material (Table S5)**. We believe this to be the first national scale assessment of possible hazard interactions in the literature, with our approach being generalizable for other national contexts. We use **Table S5** to inform the development of an additional national-scale 21×21 matrix to communicate uncertainty regarding each interaction. In **Figure 4**, blue shading indicates the number of evidence types (**A–E**) supporting the inclusion of each interaction. Darker shading indicates inclusion based on *more* evidence types and lighter shading indicates inclusion based on *less* evidence types. We group triggering and increased probability interaction types together and indicate the number of evidence types available per *primary hazard-secondary hazard* combination. This is due to the coarse resolution of the data used, and complexities of distinguishing in evidence types between triggered/increased probability interaction types. Using **Figure 4** we note that of the 50 identified interactions:

i.   2 (4%) have 5 evidence types to support their inclusion. Examples include *storm* → *landslide,* and *storm* → *flood.*

ii.  3 (6%) have 4 evidence types to support their inclusion. Examples include *landslide* → *flood,* and *storm* → *ground collapse*.

iii. 6 (12%) have 3 evidence types to support their inclusion. Examples include *earthquake* → *tsunami, landslide* → *tsunami,* and *extreme temperatures (heat)* → *wildfire.*

iv.  15 (30%) have 2 evidence types to support their inclusion. Examples include *tsunami* → *flood, drought* → *soil subsidence,* and *storm* → *ground heave.*





v.  17 (34%) have 1 evidence types to support their inclusion. Examples include *earthquake → volcanic eruption, flood → landslide,* and *storm → tsunami.*

vi.  7 (14%) are included due to globally relevant literature, rather than Guatemala-specific literature. Examples include *impact event → landslide,* and *regional subsidence → flood.*

**Figure 4** demonstrates the importance of a multi-methods approach, integrating diverse evidence types to understand relevant hazard interactions. Analysing any one evidence type (**A–E**) would only identify a sample of relevant interactions. **Table S5** shows that 13 (26%) of 50 relevant interactions were identified in the workshop of civil protection professionals (**Section 2.6**), 9 (18%) using civil protection bulletins (**Section 2.3**), 28 (56%) using interviews with hazard professionals (**Section 2.5**), and 32 (64%) using international literature (**Section 2.2**). Developing comprehensive regional interaction frameworks requires multiple, diverse evidence types.

### 3.3.2 Guatemala Southern Highlands 33×33 Interaction Framework (Matrix Form)

We now proceed to develop a regional interaction framework for a sub-national spatial extent. Using physiography, we divide Guatemala into four spatial regions (1) low relief northern plateau, (2) Central Highlands, with deep valleys, (3) Southern Highlands, and (4) Pacific coastal plains, as indicated in **Figure 5**. In **Table 9**, we show the 37 hazard sub-types described in **Section 3.2** and use the evidence in **Section 2** (**A–E**) to characterise their spatial relevance in these four regions. More hazards are spatial relevant to the Southern Highlands of Guatemala than other regions in Guatemala. 33 (89%) of 37 possible hazard sub-types are possible in the Southern Highlands of Guatemala, compared with 26 (70%) to 27 (73%) of 37 hazard sub-types relevant in the other regions. The Southern Highlands is a region of variable topography between the Pacific Coast and the Polochic-Motagua-Chamalecón fault system. It incorporates the volcanic arc, with at least three active volcanic systems (Pacaya, Fuego and Santiaguito).

The 33 hazard sub-types relevant in the Southern Highlands are used as primary and secondary hazards in our regional interaction framework. This results in 1089 (33×33) possible interactions between these hazard sub-types. Using existing global interaction frameworks (e.g., Gill and Malamud, 2014) and evidence in **Section 2**, we systematically examine each cell to determine if an interaction could or could not occur. In **Figure 6** we present this 33×33 sub-national interaction framework for the Southern Highlands of Guatemala. **Figure 6** includes 114 (10%) of 1089 cells shaded, indicating 114 possible interactions. These include:

i.  *Triggering Only.*                          26 (23%) of 114 interactions.

ii.  *Increased Probability Only.*              15 (13%) of 114 interactions.

iii.  *Triggering and Increased Probability.*    73 (64%) of 114 interactions.



The 114 interactions in **Figure 6** include interactions that occur over large and small spatial areas, with both high and low frequencies, and both high- and low-magnitude events. The temporal relevance of interactions in **Figure 6** may change, for example due to evolving anthropogenic activity (see **Section 3.5**) or environmental change. Interactions include some originating outside of the spatial region of interest, and others that may propagate outside. For example, (i) an earthquake

north of the Southern Highlands may result in ground shaking, liquefaction, landslides and other secondary hazards inside the Southern Highlands, (ii) lahars triggered in the Southern Highlands may trigger flooding outside of the Southern Highlands, in the Pacific coastal plains, and (iii) large volcanic eruptions in the Southern Highlands can eject ash/tephra far beyond this extent. Characteristics of interactions (e.g., likelihood) are not included in **Figure 6**, but could be added as additional information layers if further research results were available.

**3.4 Networks of Interacting Hazards (Cascades)**

In addition to one hazard triggering or increasing the probability of another hazard, longer linear or non-linear networks of hazard interactions (or cascades) can also occur (Han *et al.,* 2007; Choine *et al.,* 2015; Gill and Malamud, 2016; Pescaroli and Alexander, 2018). These networks include both high and low likelihood events, having diverse impacts. The civil protection bulletins characterised in **Section 2.3** include several examples of networks in Guatemala. These

include events with primary, secondary and tertiary hazards, as well as events reporting primary hazards changing the likelihood of future hazards. **Table 10** gives four diverse examples of networks derived from **Table S2**, demonstrating the complexity of networks of interacting hazards in Guatemala:

  i.    A primary hazard triggering and increasing the likelihood of multiple secondary hazards (*Example 1*).

  ii.   Linear events with primary, secondary and tertiary hazards (*Example 2*).

iii.   Multi-branch events with primary, secondary and tertiary hazards (*Example 3*).

  iv.   High-magnitude, complex, live event (tropical storm) with possible interactions (*Example 4*).

We can extract additional examples of networks from other evidence in **Section 2**. For example, stakeholder interviews (**Section 2.5**) described volcanic eruptions and heavy rain triggering lahars, which subsequently trigger floods. Networks can be visualised using interaction frameworks, as illustrated in **Figures 7** and **8**:

i.    *Case Study 1 (**Figure 7**): Lahars triggered on the flanks of Santiaguito, which result in severe erosion and trigger flooding.* This example featured in evidence in **Sections 2.2**, **2.4** and **2.5**. It occurs annually in the rainy season, while Santiaguito is active and generating large volumes of tephra.

  ii.   *Case Study 2 (**Figure 8**): Hurricane Stan (2005) triggering a debris flow in the mountains adjacent to Lake Atitlán, with this debris flow triggering a tsunami, which caused a small lakeside flood.* This example featured





in evidence in **Sections 2.2**, **2.4** and **2.5**. It occurs less frequently than Case Study 1, based on a specific event in 2005, Hurricane Stan (Luna, 2007).

The regional interaction frameworks in **Section 3.3** can help to visualise case studies of cascades identified through various evidence types. We can also use them to consider *potential* networks, given a primary event. For example, given

a large earthquake, the possible scenarios that may arise could be visualised using **Figures 3** and **6**, and evaluated by hazard professionals.

### 3.5 Anthropogenic Processes

In **Sections 3.2** to **3.4**, we primarily consider interactions between natural hazards; however, anthropogenic processes can also trigger natural hazards and influence natural hazard interactions (Glade 2003; Knapen *et al.,* 2006; Owen *et al.,* 2008;

Gill and Malamud, 2017). Information on relevant anthropogenic processes can support hazard and civil protection professionals to evaluate how anthropogenic activity may trigger hazards and influence hazard interactions.

Using a classification of 18 anthropogenic processes (Gill and Malamud, 2017), and evidence from **Section 2**, we identify 17 relevant anthropogenic processes in Guatemala, listed in **Table 11**. Some of these processes are only relevant for small spatial extents (e.g., individual towns), with others more widespread (e.g., in many populated regions). **Table 11** includes

the evidence (**A**–**E**) used to justify their relevance in Guatemala. Some anthropogenic processes feature multiple times within one evidence type. For example, four interviewees noted road construction (*Infrastructure Construction: Unloading*) and four noted deforestation (*Vegetation Removal*), in the context of triggering landslides. In contrast, only one interview participant mentioned groundwater abstraction as a potential trigger of subsidence.

The spatial and temporal relevance of these 17 anthropogenic processes will vary and could change over time.

Anthropogenic processes can start and stop, and both grow and shrink in their spatial extent. The anthropogenic processes in **Table 11** should be regularly reviewed to assess their relevance and if other processes have started, and any consequences of this variation on natural hazards and hazard interactions. For example, increased road construction may change the likelihood of landslides during heavy rain.

### 3.6 Regional Interaction Framework Summary

In this section we have integrated diverse evidence types regarding hazards and hazard interactions in Guatemala, and unified them in a formal structure, supported by expert knowledge. We have collated information on relevant single hazards and appropriate ways to classify these in Guatemala, and information on relevant hazard interactions. Using a comprehensive and systematic approach, we have constructed evidenced national and sub-national interaction frameworks in matrix form, considering networks of interacting hazards and relevant anthropogenic processes. We have

demonstrated that our approach is scalable (with national and sub-national applications described) and therefore suggest



that it is reproducible in diverse geographical contexts, and at multi-national to local scales, to generate useful insights into and assessments of hazards and hazard interactions.

## 4 Discussion

In this section, we summarise potential limitations and uncertainties within our evidence, approach, and regional interaction frameworks (**Section 4.1**). We contrast our regional interaction frameworks with in-country civil protection perspectives by using a correlation coefficient (**Section 4.2**) and discuss the operationalisation of interaction frameworks (**Section 4.3**). We conclude by discussing the development of regional interaction frameworks for additional geographical contexts (**Section 4.4**).

### 4.1 Limitations and Uncertainty.

Evidence types **A–E**, characterised in **Section 2**, are each associated with limitations and uncertainties. We note examples of these below:

i.  *Information Accuracy*. It may be difficult to verify information within grey literature sources, including media articles and textbooks (**Section 2.2**), civil protection bulletins (**Section 2.3**), and personal perspectives offered through interviews (**Section 2.5**) and workshops (**Section 2.6**). Where possible, we evaluated authenticity by cross-referencing grey and older literature with peer-review and recent literature.

ii. *Bias Towards High-Impact Events*. Civil protection bulletins (**Section 2.3**), like newspaper articles, focus on events that affect the things humans value (Carrara *et al.,* 2003), and thus exclude events with a low societal impact. In contrast to newspaper records, bulletins are less likely to focus on novel events (Moeller, 2006) and it is reasonable to expect a higher level of specialist understanding compared to newspaper journalists (Ibsen and Brunsden, 1996).

iii. *Information Omission.* Our semi-structured approach to interviews (**Section 2.5**) may make it difficult to focus on important issues (Kitchin and Tate, 2000), increasing the likelihood of missing pertinent topics.

iv. *Language Barriers.* The evidence in **Section 2** required working across language barriers. Information bulletins (**Section 2.3**) required translation from English to Spanish (when selecting keywords) and Spanish to English (when analysing keyword search results). We did not translate all text in the 677 pages of the bulletins, but rather searched for keywords within the text, and examined their context. Working in a non-native language may have resulted in missing interactions and/or misunderstanding context. Interviews and the workshop (**Sections 2.5** to **2.6**) were conducted in a non-native language (either for us or the interviewee) making it harder to ensure consistency and minimise the omission of information (Squires, 2009). The use of translators may also result in



challenges (Temple, 2002; Temple and Young, 2004). Translators can change the meaning of questions, directly or indirectly contribute to answers, or change interview dynamics. Careful selection of translators can minimise the impact of these limitations.

v.  *Cultural Barriers and Positionality.* Interviews and the workshop (**Sections 2.5** to **2.6**) involved working across cultures. Our position in social and cultural structures influences our perspective of the world, and the way that this then influences the conduct and interpretation of stakeholder engagement (e.g., Merriam *et al.,* 2001; Sultana, 2007; Fisher, 2015). Race, nationality, age, gender, social and economic status influence our positionality (Madge, 1993), as do prior experiences pertinent to this research. The interviewer, translator and interviewees may have different perspectives, value systems, customs and social behaviours. Relationships between these groups can be complex and dynamic, with similarities and differences (Merriam *et al.,* 2001). Recognising cultural differences and similarities has implications on how to manage interview contexts to ensure that they are fruitful (Schneider and Barsoux, 2002).

vi.  *Participant Selection.* Hosts at CONRED and INSIVUMEH generally selected interview and workshop participants (**Sections 2.5** to **2.6**). We desired participants from a diversity of professional backgrounds and levels of seniority, and this was generally respected. While participant selection was not in our control, the purposeful sampling used was an appropriate approach (MacDougall and Fudge, 2001; Longhurst, 2003; Suri, 2011; Palinkas *et al.,* 2015).

vii.  *Power Dynamics.* Age, gender, educational level, ethnicity and socio-economic status can influence an interview or workshop (**Sections 2.5** to **2.6**) process and the results (e.g., Valentine, 1997; Edwards, 1998; Kitchin and Tate, 2000; Qu and Dumay, 2011). Genuine rapport, respect, trust, and an understanding of cultural differences can reduce the impact of power dynamics (Kitchin and Tate, 2000; DiCicco-Bloom and Crabtree, 2006).

viii.  *Peer Influence.* During the workshop (**Section 2.6**), a controlled environment was encouraged during the completion of tasks. It was, however, difficult to prevent those sitting next to each other from seeing other contributions and speaking about what they were including.

ix.  *Hazards and Interaction Classifications.* Gill and Malamud (2016) discussed difficulties in distinguishing between triggering and increased probability interaction types. Workshop participants (**Section 2.6**) may have found this distinction confusing, or defined interaction types and hazard classifications in different ways.

These examples are likely to have resulted in some uncertainty within the evidence used, and therefore within the interaction frameworks produced using this evidence. Some sources of uncertainty can be mitigated, and appropriate actions were taken to do so. For example, a reflexive and respectful approach can reduce language barriers, cultural barriers and power dynamics on the results of stakeholder engagement, and a critical approach to literature analysis can



determine where inaccuracies may exist in grey or historical literature. Integrating multiple evidence types also helps to reduce the impact of uncertainties on regional interaction frameworks. We can cross-reference personal perspectives expressed in interviews, for example, with peer-review literature to explore accuracy. Global interaction frameworks also serve as useful databases of what could occur, helping to evaluate the scope of possible interactions before ascertaining

their relevance in Guatemala. We suggest, therefore, that the regional interaction frameworks presented in **Section 3** are robust assessments of potential triggering and increased probability interactions in Guatemala. It is possible, however, that relevant hazard interactions and anthropogenic processes, or the likelihood or spatial distribution of these, will vary over time.

In addition to the evidence used to populate our regional interaction frameworks, their form (two-parameter matrices)

may result in some complex interaction types not being captured. For example, two or more independent hazards may coincide spatially and/or temporally and result in a complex network of hazard interactions. Information in the frameworks could be represented spatially to help examine such scenarios. Two or more independent hazards may also trigger other hazards (e.g., storm and volcanic eruption triggering lahars). While the national framework (**Figure 3**) does not capture this example, the sub-national framework (**Figure 6**) with the expanded hazard classification does capture it

(illustrated in **Figure 7**). Non-linear examples can be visualised in this way.

## 4.2 Collective Knowledge of Hazard Interactions

Hazard interactions cut across multiple disciplines and so require input from diverse specialisms (Kappes *et al.,* 2012; Scolobig *et al.,* 2013; Scolobig *et al.,* 2017). Interaction frameworks could therefore help to facilitate enhanced cross-institutional dialogue about hazard interactions, their likelihoods and potential impacts. This could help to strengthen

collective knowledge of hazard interactions, and the ability of an individual to access this knowledge. By contrasting results from our workshop (**Figure 2**) with our Guatemala national interaction framework (**Figure 3**), we can examine and quantify congruence between the two matrices. **Figure 9** is a 21×21 interaction matrix that combines **Figures 2** and **3** to indicate the number of workshop participants (from a total of 16) that identified an interaction as being relevant to Guatemala (numbers), and the interactions identified within our national interaction framework (grey shading, from

**Figure 3**).

**Figure 9** combines information and knowledge from 16 participants to present something that is 'owned' by no individual. It is collective knowledge, combining information and knowledge owned by multiple people (Antonelli, 2000). We do not expect an individual scientist or hazard professional to map out all relevant interactions. Assessing how an organisation rather than an individual understands interactions demonstrates their collective knowledge. For this

knowledge to be truly collective there must be effective communication between participants, and a means by which this knowledge can be accessed, shared and applied (Foray, 2000; Antonelli, 2000; Paton *et al.,* 2008).





Multi-hazard research is complex, and requires scientists and professionals operating in many different disciplines. **Figure 9** demonstrates large variation in perspectives between participants on hazard interactions There is a unanimous consensus (i.e., 16 participants) that an interaction exists in two (0.5%) of 441 possible triggering interactions. To assess congruence between the participants' perspectives (numbers in **Figure 9**) and national interaction framework (grey shading in **Figure 9**), we use Matthews' Correlation Coefficient, or *MCC* (Matthews, 1975). *MCC* values are a function of true positives (TP), true negatives (TN), false positives (FP), and false negatives (FN) and can be expressed as follows (Matthews, 1975; Powers, 2011):

$$MCC = \frac{TP \times TN - FP \times FN}{\sqrt{(TP + FP)(TP + FN)(TN + FP)(TN + FN)}}$$ (**Equation 1**)

The *MCC* gives a value of congruence between '−1.0' (zero overlap between the numbers and grey shading in **Figure 9**) and '+1.0' (perfect overlap between the numbers and grey shading in **Figure 9**). An *MCC* = 0.0 suggests that the amount of congruence is no better than a random average (Kaufmann *et al.,* 2012). We use two different approaches:

    i.   *All identified interactions*. Where ≥1 people note an interaction to be relevant, we consider this part of the group's collective knowledge. From **Figure 9** we identify 86 interactions identified by the 16 workshop participants. This compares to 50 interactions in the national framework, **Figure 3**.

    ii.   *Interactions identified by '≥ x' participants.* A threshold could be applied, in terms of the number of participants identifying a given natural hazard interaction. Only those interactions that reach or exceed this threshold are considered. We select thresholds of ≥3 and ≥5 (out of 16 workshop participants) identifying an interaction as being relevant. From **Figure 9** we identify 32 and 19 possible interactions for these respective scenarios. These thresholds demonstrate a method for considering what constitutes collective knowledge, but others could be selected.

Using three thresholds (≥1, ≥3, ≥5), we calculate Matthews' Correlation Coefficients (*MCC*) using **Equation 1**. These are presented in **Table 12** and are *MCC* = 0.28 when all interactions are considered (≥1), improving to *MCC* = 0.51 with a threshold of ≥3 participants and *MCC* = 0.49 with a threshold of ≥5 participants noting an interaction. Applying a threshold of ≥3 (vs. ≥1) people identifying an interaction has a slight influence on the number of true positives (22 vs. 24 interactions) but significantly reduces the number of false positives (10 vs. 62 interactions). Using a sensitivity test, where the number of TP and TN are varied by +1, *MCC* changes by 0.02 for each additional TP and 0.01 for each additional TN. For example, a participant identifying 12TP and 374TN will have an MCC = 0.25, whereas a participant identifying 13TP and 375TN will have an MCC = 0.28 (=0.25+0.01+0.02).





*Matthews' Correlation Coefficient* is a simple indicator of agreement, which we use to examine differences between stakeholder perspectives and our national interaction framework (**Figure 3**). When applying a small threshold (≥3 people agreeing on a given interaction) to determine which interactions were analysed, the collective knowledge of 16 participants generated the closest agreement to the national interaction framework (*MCC* = 0.51). This *MCC* is based on

22 (44%) of 50 interactions in **Figure 3** being identified by ≥3 participants, and therefore 28 (56%) of 50 interactions that ≤2 participants identified in the workshop. Of these 27 interactions identified by ≤2 participants, nobody identified 25 different interactions. These results suggest the following:

- *Enhanced communication within and across organisations involved in natural hazards and DRR in Guatemala could help when considering hazard interactions.* Interaction frameworks could help facilitate this communication and

elicit additional information about interaction likelihoods and impacts. Ensuring that collective understanding of hazard interactions is operationalised to greatest effect will require strong institutions, and cross-departmental and cross-disciplinary communication (Scolobig *et al.,* 2017).

- *National and sub-national interaction frameworks could promote dialogue on both high- and low-likelihood events.* Interactions in the national interaction framework (**Figure 3**) include some low-likelihood hazard interactions, such

as impact events triggering tsunamis, and storms triggering (meteo)tsunamis. Workshop participants may not consider low-likelihood events due to lack of access to peer-review literature. Only 5 of the 21 interview participants (**Section 2.5**) had access to, or regularly used, peer-review journals. Interview participants predominantly relied on experience and communication with colleagues for further information on natural hazards and interactions.

- *We can use MCC values to monitor changing understanding and perceptions of natural hazard interactions. MCC*

values can be determined before interaction frameworks are introduced into an organisation, and then recalculated weeks, months, or years after individuals have explored, discussed and used them in their work.

### 4.3 Operationalisation of Interaction Frameworks

Engagement with hazard and civil protection professionals, academics, the private sector and intergovernmental organisations informed our development of regional interaction frameworks for Guatemala. Understanding stakeholder

requirements (e.g., terminology, spatial scales and temporal scales) can ensure frameworks are fit-for-purpose. In 2018, we returned to Guatemala and shared our interaction frameworks through seminars, roundtable discussions and interviews. We elicited perspectives on (i) the structure and content of the interaction frameworks, (ii) use of the interaction frameworks, and (iii) future research and innovation opportunities. This engagement highlighted some common themes:





- **Understanding Multi-Hazard Interactions.** The interaction frameworks are a visual synthesis of diverse knowledge, traditionally 'owned' by diverse disciplinary groups. This can help to enhance awareness of the spectrum of hazards and hazard interactions in a given territory, and strengthen communication across disciplinary boundaries. Interaction frameworks allow those undertaking research into any particular single hazard to place their work within

the context of other natural hazards, thus fostering communication between hazard specialists and encouraging a more interdisciplinary approach. One participant noted that '*sometimes knowledge is in a head, but now it is in a visual summary* [that can be used by a range of people]'. Furthermore, some participants questioned the inclusion of particular hazards and/or hazard interactions in the interaction frameworks. Examples include, earthquakes triggering volcanic eruptions, floods triggering volcanic eruptions, landslides triggering tsunamis. Following discussion of the

evidence, participants reported changes in opinion about their relevance and inclusion within planning.

- **Multi-Hazard Research.** Additional information layers (e.g., thresholds, likelihoods, scales of impact) could inform decision making around natural hazards. This requires new research to understand hazard and disaster dynamics in Guatemala. A 'multi-hazards' observatory could enable the collection of diverse data to better characterise these layers of information. Forensic studies of past and ongoing disasters, using interdisciplinary approaches, would

generate new insights into potential impacts.

- **Scales of Interest.** Many participants suggested that municipalities are the preferred scale of interest for further multi-hazard tools. Guatemala currently has 340 municipalities, across 22 Departments. The emphasis on municipalities likely arises from the political context in Guatemala, with municipal authorities being the final users of information. Other stakeholders noted that it may not be most effective (or efficient) to produce municipal-scale hazard

assessments as hazards cross municipal, departmental, and national boundaries (Gill, 2016). Tools can be prepared at scales that both provide useful information to those working at a municipal scale and recognise the artificial nature of these boundaries.

- **Preferred Tools and Technologies.** Participants were interested in tools allowing the spatial representation of information in **Section 3**. A GIS tool allowing the creation of municipal multi-hazard risk maps was a high priority,

allowing the identification of hazard hotspots, improved disaster preparation (e.g., evacuation routes), and enhanced response through improved communication of potential secondary hazards. Spatial representation of information could help to identify regions where secondary hazards are more likely after a primary hazard, and the assessment of disaster impacts, including those generated through secondary hazards, by overlay of exposure and multi-hazard maps.

In addition to these generalised themes relating to next steps, participants noted specific ways that they could use our hazard interaction frameworks in their ongoing work. INSIVUMEH, CONRED and UN-OCHA indicated that they could





use interaction frameworks as reference tools to strengthen preparedness and response to hazards. CONRED suggested they could integrate secondary hazards information into their public information bulletins and requested blank matrices to complete for specific high-risk municipalities. Finally, universities indicated that they would use this research and our systematic classification of hazards in Guatemala in their teaching. Fully realising the impact of regional interaction frameworks, and ensuring positive social impact, will require sustained collaborative engagement with user communities. The potential developments and applications outlined above would support the embedding and operationalisation of this research in Guatemala and may prove insightful in other settings.

### 4.4 Application of Methods to Other Contexts

Our approach, set out through **Sections 1** to **3**, is scalable and can be applied in diverse geographical contexts. A synthesis of available evidence in any given context (e.g., multi-national, national, sub-national) is necessary to underpin the construction of regional interaction frameworks. This process, outlined in **Section 3**, first develops a location-specific hazard classification, and then populates a customised matrix with information about relevant hazard interactions. Other countries in Central America (e.g., Nicaragua, El Salvador, Costa Rica) have similarities to Guatemala in their hazard landscape. Their national interaction frameworks would likely be similar, although not identical, to Guatemala. Interaction frameworks for other countries may look very different, shaped by the tectonic and meteorological setting.

Regional interaction frameworks can also be developed for sub-national scales, including large geographical domains, municipalities, or localised sites important to the development of critical infrastructure. We propose that comprehensive, systematic and evidenced regional interaction frameworks can be developed for and operationalised in diverse settings to improve DRR and response. When co-created by diverse stakeholders, interaction frameworks can help to facilitate communication across specialisms engaged in hazard monitoring and civil protection. Through dialogue, it may be possible to further characterise interactions, identifying those that are most likely to occur and those that could cause the greatest damage. This may help to improve decision making in key agencies engaged in DRR and civil protection.

### 5 Conclusions

In this paper, we have described an approach to develop comprehensive, systematic and evidenced regional interaction frameworks that inform multi-hazard approaches to DRR, as encouraged by the Sendai Framework for DRR. We have applied this methodology to Guatemala, presenting regional interaction frameworks for the national spatial extent of Guatemala and sub-national spatial extent of the Southern Highlands of Guatemala. Five evidence types (internationally accessible publications and reports, locally accessible civil protection bulletins, field observations, semi-structured stakeholder interviews, and a stakeholder workshop) underpin these frameworks, and we use this evidence to do the following:





i.   *Determine an appropriate classification scheme.* For Guatemala, this consists of six natural hazard groups, 19 hazard types, and 37 hazard sub-types.

ii.  *Identify potential natural hazard interactions.* For a national spatial extent in Guatemala, we identify 50 possible interactions between 19 relevant natural hazard types. For the Southern Highlands of Guatemala, we identify
114 possible interactions between 33 relevant natural hazard sub-types.

We present information in accessible visualisations, primarily interaction matrices. The use of accessible visualisation tools, such as matrices, to represent complex hazard interactions contributes to knowledge exchange across different disciplines. The national and regional multi-hazard interaction frameworks presented here are communication tools that can enhance the application of multi-hazard research and collective knowledge in DRR management and policy. We also
consider potential networks of hazard interactions, and constrain relevant anthropogenic processes using the evidence outlined above. Our approach allows those working on any individual hazard in Guatemala to place their work within the context of other natural hazards. When taking our regional interaction frameworks back to Guatemala, we observed them fostering communication between hazard specialists and encouraging integrated multi-hazard approaches to DRR.

We believe our approach is scalable and can be replicated in diverse geographical settings. While examples of regional
interaction frameworks exist in the literature, these often do not include a systematic assessment of possible natural hazards and interactions for a defined spatial extent. By integrating diverse evidence types, we have developed a systematic approach that constrains relevant interactions between a comprehensive selection of natural hazards. We simplify a broad array of complex information to facilitate an effective analysis by those working on reducing and managing the risk from natural hazards within both policy and practitioner sectors.

**Acknowledgements**

The lead author initiated this work while a PhD candidate in the Department of Geography, King's College London, funded by a NERC/ESRC studentship (grant: NE/J500306/1). Subsequent engagement in Guatemala was funded by the British Geological Survey Innovation Flexible Fund, and supported by BGS NC-ODA grant NE/R000069/1: Geoscience for Sustainable Futures. We are grateful to staff at INSIVUMEH, CONRED, and USAC for their engagement with our
work. We are particularly thankful to Gustavo Chigna (INSIVUMEH) for his sustained advice and support in the field while in Guatemala. We thank Roxana Ciurean, ###### and ###### for their constructive reviews. This article is published with the permission of the Executive Director, British Geological Survey (UKRI).




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



# Figures and Tables

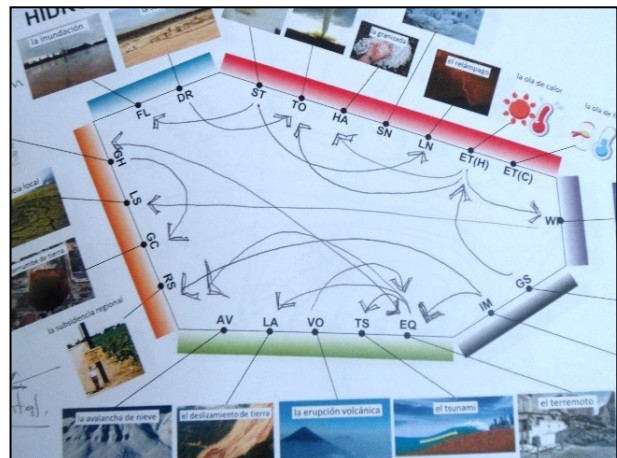
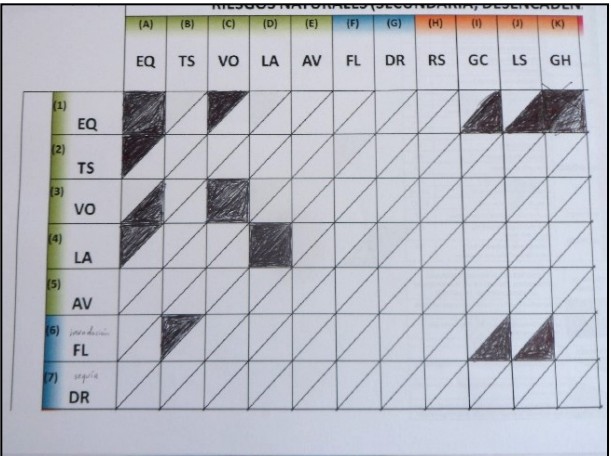

**Figure 1. Stakeholder identification of possible hazard interactions in Guatemala.** Two examples of visual records collected in Guatemala, including (A) a network linkage diagram for 21 hazards, and (B) a 7 × 11 hazard interaction matrix. Both were completed during a workshop in Guatemala on 6 March 2014. The workshop is described in **Section 2.6**, and all images from the workshop (16 network linkage diagrams, and 15 hazard interaction matrices) are included in the **Supplementary Material (Figures S1** and **S2)**.





Natural Hazards and Earth System Sciences — Open Access — EGU — Discussions

**SECONDARY HAZARD (TRIGGERED OR INCREASED PROBABILITY)**

| PRIMARY HAZARD | (A) EQ | (B) TS | (C) VO | (D) LA | (E) AV | (F) FL | (G) DR | (H) RS | (I) GC | (J) SS | (K) GH | (L) ST | (M) TO | (N) HA | (O) SN | (P) LN | (Q) ET(H) | (R) ET(C) | (S) WF | (T) GS | (U) IM |
|---|---|---|---|---|---|---|---|---|---|---|---|---|---|---|---|---|---|---|---|---|---|
| (1) EQ | 5 | 11 | 4 | 16 | 2 | 2 | | 7 | 8 | 8 | 13 | 1 | | | | | | | 1 | | |
| (2) TS | | | | | | 4 | | | | | | | | | | | | | | | |
| (3) VO | 8 | 2 | | 7 | 1 | | | 2 | 1 | 1 | 1 | | | | | | | | 4 | | |
| (4) LA | | | | 1 | | | | 1 | 1 | 1 | 3 | | | | | | | | | | |
| (5) AV | | | | | | | | | | | | | | | | | | | | | |
| (6) FL | | | | | | | 1 | | 1 | 1 | | | | | | | | | | | |
| (7) DR | | | | | | | | 1 | | | | | 1 | | | | | | 5 | | |
| (8) RS | | | | | | | | | 1 | 1 | 1 | | | | | | | | | | |
| (9) GC | | | | | | | 1 | 1 | | 1 | 2 | | | | | | | | | | |
| (10) SS | | | | | | | | 2 | 1 | | 1 | | | | | | | | 1 | | |
| (11) GH | 2 | | | | | | | 1 | 2 | 2 | | | | | | | | | | | |
| (12) ST | | | | 12 | 1 | 16 | | 3 | 9 | 5 | 5 | | | 3 | | 4 | | | | | |
| (13) TO | | | | | | | 1 | | | | | 1 | | | | 1 | | | | | |
| (14) HA | | | | 1 | | | | | | | | 1 | | | | | | | | | |
| (15) SN | | | | 1 | | | | | | | | 1 | | | | | | | | | |
| (16) LN | | | | | | | | | | | | | | | | | | | 8 | | |
| (17) ET (H) | | | | | | | 12 | | | | | 1 | 1 | | | | | | 11 | | |
| (18) ET (C) | | | | 1 | | | | | | | | 1 | 1 | 6 | 4 | | | | | | |
| (19) WF | | | | 1 | | 3 | | | | | | | | | | | | | | | |
| (20) GS | | | | | | | | | | | | | | | | | 3 | | 2 | | |
| (21) IM | 3 | | | 1 | | | | 3 | 1 | 1 | 2 | | | | | | | | 3 | | |

**KEY**

| HAZARD GROUP | HAZARD | CODE |
|---|---|---|
| GEOPHYSICAL | Earthquake | EQ |
| | Tsunami | TS |
| | Volcanic Eruption | VO |
| | Landslide | LA |
| | Snow Avalanche (not relevant) | AV |
| HYDROLOGICAL | Flood | FL |
| | Drought | DR |
| SHALLOW EARTH PROCESSES | Regional Subsidence | RS |
| | Ground Collapse | GC |
| | Soil (Local) Subsidence | SS |
| | Ground Heave | GH |
| ATMOSPHERIC | Storm | ST |
| | Tornado | TO |
| | Hailstorm | HA |
| | Snowstorm (not relevant) | SN |
| | Lightning | LN |
| | Extreme Temperature (Hot) | ET (H) |
| | Extreme Temperature (Cold) | ET (C) |
| BIOPHYSICAL | Wildfire | WF |
| SPACE | Geomagnetic Storm | GS |
| | Impact Event | IM |

| SYMBOL | EXPLANATION |
|---|---|
| 12 | Number of workshop participants (n=16) identifying interactions as relevant in Guatemala using hazard linkage diagrams (example below) |

**Figure 2. Stakeholder identification of possible hazard interactions in Guatemala, using network linkage diagrams produced by 16 civil protection professionals in Guatemala.** A 21 × 21 matrix with primary hazards on the vertical axis and secondary hazards on the horizontal axis. These hazards are coded, as explained in the key. Numbers indicate the number of stakeholders (from a maximum of 16) proposing each hazard interaction as being possible in Guatemala.





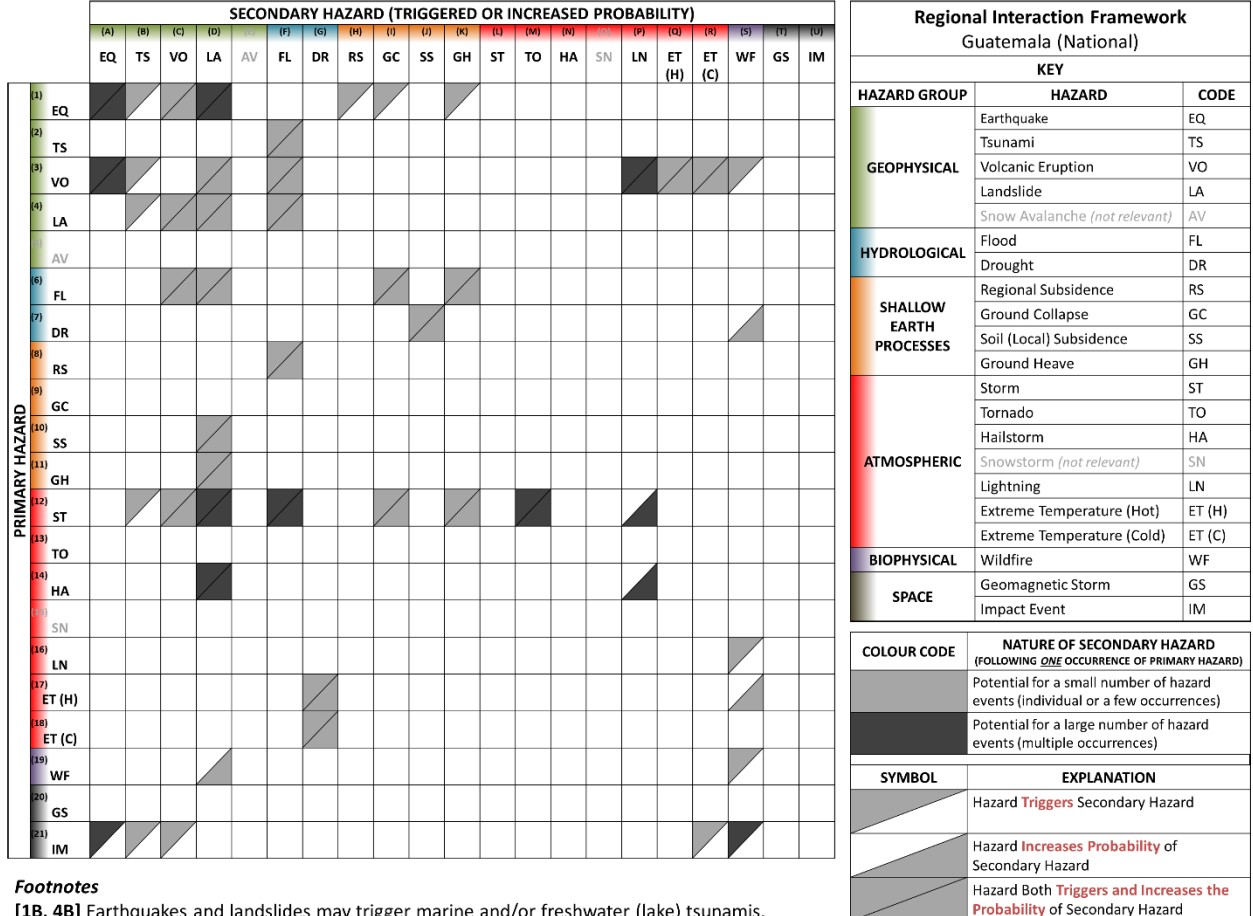

**Footnotes**

[1B, 4B] Earthquakes and landslides may trigger marine and/or freshwater (lake) tsunamis.

[1C,H; 12M] There was uncertainty about the nature of these relationships.

[1I,K] Earthquakes may trigger collapse/heave primarily through liquefaction.

[3B] Volcanic explosions may trigger freshwater tsunamis in the Lakes of Guatemala.

[3Q/R] Volcanic eruptions can trigger temperature changes if they are of sufficient magnitude.

[6,12C] Water input triggers or increases the probability of a phreatic/phreatomagmatic eruption.

[8F] Although regional subsidence triggering flooding was not noted in any evidence sources consulted, this is an inevitable consequence of the lowering of the ground surface.

[12B] Pressure changes associated with storms may trigger meteotsunamis in marine environments.

[21A-C,R,S] Identified as being generally possible, supported by globally-relevant literature rather than location-specific evidence.

**Figure 3. National Interaction Framework for Guatemala.** A 21×21 matrix with 21 primary natural hazards on the vertical axis, and 21 secondary natural hazards on the horizontal axis. Interactions (shaded cells) include primary hazards triggering a secondary hazard, and primary hazards increasing the probability of a secondary hazard. This matrix is populated using different evidence types, as outlined through **Section 2**. Visualisation structure based on Gill and Malamud (2014).





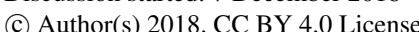

**Footnotes**

**[1B, 4B]** Earthquakes and landslides may trigger marine and/or freshwater (lake) tsunamis.

**[1C,H; 12M]** There was uncertainty about the nature of these relationships.

**[1I,K]** Earthquakes may trigger collapse/heave primarily through liquefaction.

**[3B]** Volcanic explosions may trigger freshwater tsunamis in the Lakes of Guatemala.

**[3Q/R]** Volcanic eruptions can trigger temperature changes if they are of sufficient magnitude.

**[6,12C]** Water input triggers or increases the probability of a phreatic/phreatomagmatic eruption.

**[8F]** Although regional subsidence triggering flooding was not noted in any evidence sources consulted, this is an inevitable consequence of the lowering of the ground surface.

**[12B]** Pressure changes associated with storms may trigger meteotsunamis in marine environments.

**[21A-C,R,S]** Identified as being generally possible, supported by globally-relevant literature rather than location-specific evidence.

**Figure 4. Evidence types used in the construction of a National Interaction Framework for Guatemala.** A 21×21 matrix with 21 primary natural hazards on the vertical axis, and 21 secondary natural hazards on the horizontal axis. Interactions (shaded cells) include primary hazards triggering a secondary hazard, and primary hazards increasing the probability of a secondary hazard. This matrix is populated using different evidence types, as outlined through **Section 2**. Blue shading indicates the number of evidence types used to populate each matrix cell, as described in the key. The coarse resolution of the data used, and complexities of distinguishing between triggered/increased probability interaction types, means we group both interaction types together when indicating the number of evidence types. Visualisation structure based on Gill and Malamud (2014).





**Figure 5. Guatemala Map: Key Locations and Physiography (CIA, 2001).** A combined political and physiographic map of
Guatemala, showing differential relief (greyscale shading), Departmental boundaries (green lines and text), key locations (black text),
rivers (blue lines and text) and roads (red lines). The Southern Highlands are also labelled (referred to throughout this paper.



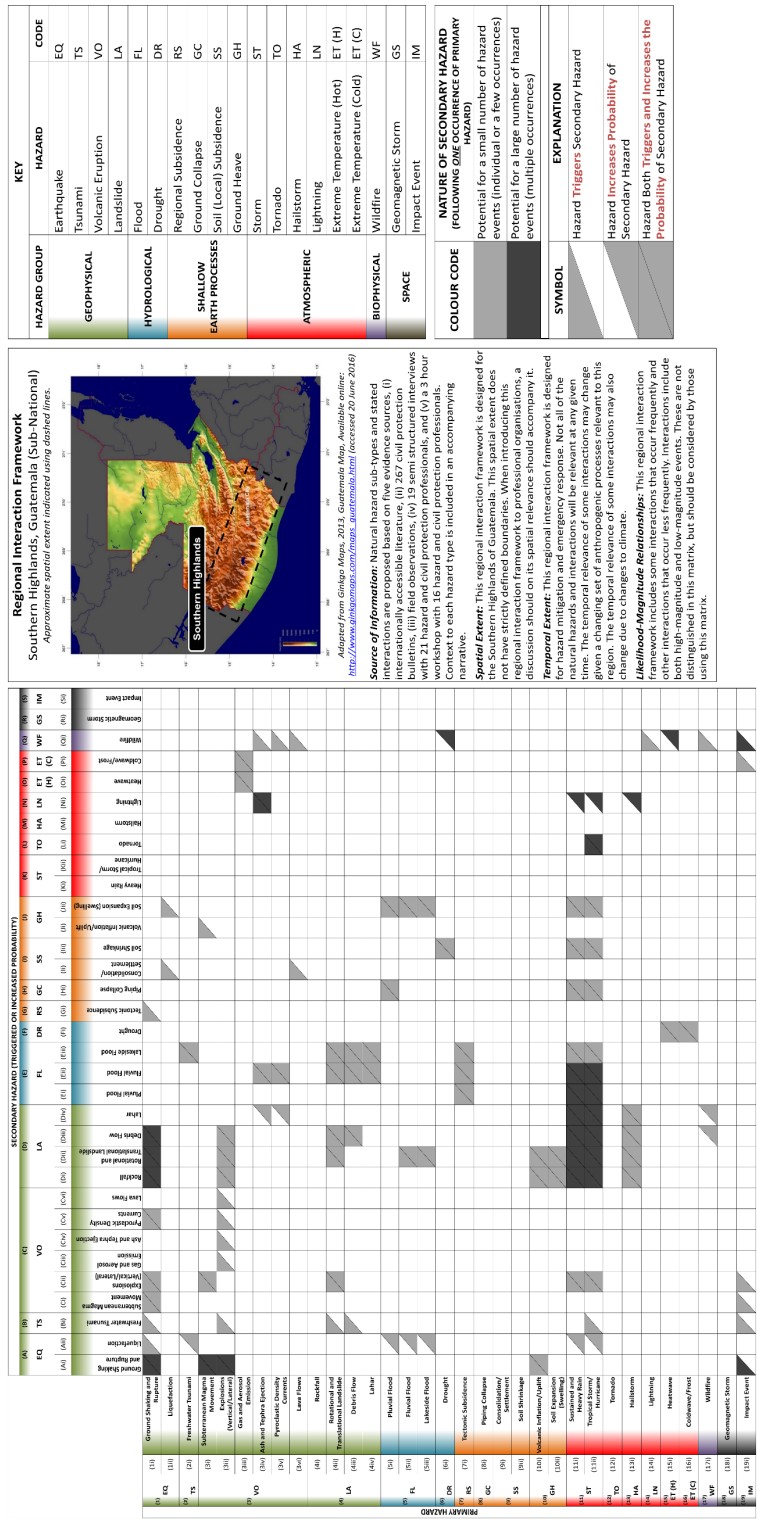

**Figure 6. Southern Highlands (Sub-National) Interaction Framework, Guatemala.** A 33×33 matrix with 33 primary natural hazard sub-types on the vertical axis, and 33 secondary natural hazard sub-types on the horizontal axis. Interactions (shaded cells) include primary hazards triggering a secondary hazard, and primary hazards increasing the probability of a secondary hazard. This matrix is populated using different evidence types, as outlined through **Section 2**.



**Figure 7. Network of Hazard Interactions (Example 1), Southern Highlands, Guatemala**. A 26 × 17 extract of the 33×33 sub-national interaction framework presented in **Figure 6**, with an example of a network of hazard interactions (cascade). This example shows (i) volcanic explosions triggering the ejection of ash and tephra, (ii) ash and tephra increasing the likelihood of lahars, (iii) heavy rain (together with the existing tephra and ash) combining to trigger a lahar. (iv) lahars triggering flooding. Evidence for this network is stated in the text.







**Figure 8. Network of Hazard Interactions (Example 2), Southern Highlands, Guatemala**. A 26 × 17 extract of the 33×33 sub-national interaction framework presented in **Figure 6**, with an example of a network of hazard interactions (cascade). This example shows (i) Hurricane Stan triggering a debris flow, (ii) debris flows triggering a freshwater tsunami in Lake Atitlan, and (iii) freshwater tsunami triggering a lakeside flood. Evidence for this network is stated in the text.





Natural Hazards and Earth System Sciences
EGU
Discussions

SECONDARY HAZARD (TRIGGERED OR INCREASED PROBABILITY)

| PRIMARY HAZARD | (A) EQ | (B) TS | (C) VO | (D) LA | (E) AV | (F) FL | (G) DR | (H) RS | (I) GC | (J) SS | (K) GH | (L) ST | (M) TO | (N) HA | (O) SN | (P) LN | (Q) ET (H) | (R) ET (C) | (S) WF | (T) GS | (U) IM |
|---|---|---|---|---|---|---|---|---|---|---|---|---|---|---|---|---|---|---|---|---|---|
| (1) EQ | 5 | 11 | 4 | 16 | 2 | 2 | | 7 | 8 | 8 | 13 | 1 | | | | | | | 1 | | |
| (2) TS | | | | | | 4 | | | | | | | | | | | | | | | |
| (3) VO | 8 | 2 | | 7 | 1 | | | 2 | 1 | 1 | 1 | | | | | | | | 4 | | |
| (4) LA | | | | | 1 | | | 1 | 1 | 1 | 3 | | | | | | | | | | |
| (5) AV | | | | | | | | | | | | | | | | | | | | | |
| (6) FL | | | | | | | 1 | | | 1 | 1 | | | | | | | | | | |
| (7) DR | | | | | | 1 | | 1 | | | | | 1 | | | | | | 5 | | |
| (8) RS | | | | | | | | | 1 | 1 | 1 | | | | | | | | | | |
| (9) GC | | | | | | 1 | | 1 | | 1 | 2 | | | | | | | | | | |
| (10) SS | | | | | | | | 2 | 1 | | 1 | | | | | | | | 1 | | |
| (11) GH | 2 | | | | | | | 1 | 2 | 2 | | | | | | | | | | | |
| (12) ST | | | | 12 | 1 | 16 | | 3 | 9 | 5 | 5 | | | 3 | | 4 | | | | | |
| (13) TO | | | | | | 1 | | | | | | 1 | | | | 1 | | | | | |
| (14) HA | | | | | 1 | | | | | | | 1 | | | | | | | | | |
| (15) SN | | | | | 1 | | | | | | | 1 | | | | | | | | | |
| (16) LN | | | | | | | | | | | | | | | | | | | 8 | | |
| (17) ET (H) | | | | | | | 12 | | | | | | 1 | 1 | | | | | 11 | | |
| (18) ET (C) | | | | | 1 | | | | | | | 1 | 1 | 6 | 4 | | | | | | |
| (19) WF | | | | 1 | | | 3 | | | | | | | | | | | | | | |
| (20) GS | | | | | | | | | | | | | | | | | 3 | | 2 | | |
| (21) IM | 3 | | | 1 | | | | 3 | 1 | 1 | 2 | | | | | | | | 3 | | |

**Regional Interaction Framework**
Guatemala (National)

KEY

| HAZARD GROUP | HAZARD | CODE |
|---|---|---|
| GEOPHYSICAL | Earthquake | EQ |
| | Tsunami | TS |
| | Volcanic Eruption | VO |
| | Landslide | LA |
| | Snow Avalanche *(not relevant)* | AV |
| HYDROLOGICAL | Flood | FL |
| | Drought | DR |
| SHALLOW EARTH PROCESSES | Regional Subsidence | RS |
| | Ground Collapse | GC |
| | Soil (Local) Subsidence | SS |
| | Ground Heave | GH |
| ATMOSPHERIC | Storm | ST |
| | Tornado | TO |
| | Hailstorm | HA |
| | Snowstorm *(not relevant)* | SN |
| | Lightning | LN |
| | Extreme Temperature (Hot) | ET (H) |
| | Extreme Temperature (Cold) | ET (C) |
| BIOPHYSICAL | Wildfire | WF |
| SPACE | Geomagnetic Storm | GS |
| | Impact Event | IM |

| COLOUR CODE | NATURE OF SECONDARY HAZARD (FOLLOWING *ONE* OCCURRENCE OF PRIMARY HAZARD) |
|---|---|
| | Potential for a small number of hazard events (individual or a few occurrences) |
| | Potential for a large number of hazard events (multiple occurrences) |

| SYMBOL | EXPLANATION |
|---|---|
| | Hazard Triggers Secondary Hazard |
| | Hazard Increases Probability of Secondary Hazard |
| | Hazard Both Triggers and Increases the Probability of Secondary Hazard |

***Footnotes***

**[1B, 4B]** Earthquakes and landslides may trigger marine and/or freshwater (lake) tsunamis.
**[1C,H; 12M]** There was uncertainty about the nature of these relationships.
**[1I,K]** Earthquakes may trigger collapse/heave primarily through liquefaction.
**[3B]** Volcanic explosions may trigger freshwater tsunamis in the Lakes of Guatemala.
**[3Q/R]** Volcanic eruptions can trigger temperature changes if they are of sufficient magnitude.
**[6,12C]** Water input triggers or increases the probability of a phreatic/phreatomagmatic eruption.
**[8F]** Although regional subsidence triggering flooding was not noted in any evidence sources consulted, this is an inevitable consequence of the lowering of the ground surface.
**[12B]** Pressure changes associated with storms may trigger meteotsunamis in marine environments.
**[21A-C,R,S]** Identified as being generally possible, supported by globally-relevant literature rather than location-specific evidence.

**Figure 9. Stakeholder identification of possible hazard interactions in Guatemala, overlain over the national interaction framework developed in Figure 3.** A 21×21 matrix with primary hazards on the vertical axis and secondary hazards on the horizontal axis. These hazards are coded, as explained in the key. These matrices show cases where a primary hazard could trigger and/or increase the probability of a secondary hazard. Grey cell shading indicates the interaction was identified in the national hazard interaction matrix presented in Figure 3. Numbers indicate the total number (from a maximum of 16) of stakeholders proposing each hazard interaction as being possible in Guatemala.




**Table 1. Examples of regional interaction frameworks. Summaries of seven regional interaction frameworks.**

| Authors | Summary<br>*(Spatial extent, hazards and processes considered, and interaction types)* |
|---|---|
| Tarvainen *et al.* (2006) | • **Continental** spatial extent (Europe).<br>• **Binary matrix**.<br>• Identifies interactions between **eleven natural hazards** (*avalanche, drought, earthquake, extreme temperature, flood, forest fire, landslide, storm surge, tsunami, volcanic eruption, winter storm*) and **four technological hazards** (*air traffic accident, chemical plant, nuclear power plant, oil processing/ transport/ storage*).<br>• Interactions are determined based on physical processes (**causal correlation**), and are only considered when hazard intensities in a given region exceed an average value. |
| De Pippo *et al.* (2008) | • **Sub-national** spatial extent (Northern Campanian coast, Italy).<br>• **Descriptive matrix** is used **to characterise interactions** between hazards, which are weighted according to their importance in different zones along the coast.<br>• **Semi-quantitative method** to quantify, rank and map the distribution of hazard.<br>• Considers the effect of **six hazards** (*shoreline erosion, riverine flooding, surge, landslide, seismicity and volcanism*) and the effect of manufactured structures. |
| Kappes *et al.* (2010) | • **Sub-national** spatial extent (French Alpine region of Barcelonnette).<br>• Uses a combination of **binary** and **descriptive matrices**.<br>• Considers both **triggering interactions** and **interactions where a hazard changes the disposition** or general setting that favours another hazard process.<br>• **Seven primary natural hazards** (*avalanche, debris flow, rock fall, landslide, flood, heavy rainfall, and earthquake*). |
| van Westen *et al.* (2014) | • **Sub-National** (European mountainous environments)<br>• Possible interactions are mapped out using a **network flow diagram**, including interactions between the seven resulting (secondary hazards). Considers **two primary triggers** (*earthquake, meteorological extremes*), and **seven resulting natural hazards** (*mass movement, snow avalanche, forest fire, land degradation, flooding, seiche, technological hazard*). |
| Neri *et al.* (2008) | • **Sub-National** (Vesuvius volcano, Italy).<br>• Uses a **quantitative (probabilistic) approach** to map out possible future eruptive scenarios.<br>• Scenarios consider **ten hazards** (*volcanic eruption, fallout, ballistics, pyroclastic density current, debris avalanche, tsunami, flood, landslide, lahar, mudslide, heavy rain*). |
| Neri *et al.* (2013) | • **Sub-National** (Kanlaon volcano, Philippines)<br>• Presented using an **event/scenario tree**.<br>• Uses a **semi-quantitative method**, combing geological and historical data to consider hazard events.<br>• **Eight hazards considered** (*volcanic eruption, fallout, ballistics, pyroclastic density current, debris avalanche, tsunami, flood, lahar/mudslide*). |
| Liu *et al.* (2016) | • **Sub-National** (Yangtze River Delta, China).<br>• **Zones** of similar hazards and hazard interactions are identified and **spatially mapped**.<br>• Hazard interactions classification is **based on the 'the hazard-forming environment'**, defined as the geophysical environment that natural hazards arise from.<br>• **Four interactions types** are considered<br>• **Ten natural hazards** (*earthquakes, volcanic eruption, tropical cyclone, slow riverine flood, fast riverine flood, coastal flood, pluvial flood, landslide, avalanche, drought*), with a selection of these being relevant to the Yangtze River Delta case study. |



**Table 2. Government organisations contributing to DRR in Guatemala.** All information taken from their respective websites (CONRED, 2018a; INSIVUMEH, 2018).

| Acronym | Full Name | Organisational Remit |
|---|---|---|
| CONRED | Coordinadora Nacional para la Reducción de Desastres *(National Coordinator for Disaster Reduction)* | Established in 1996 and responsible for preventing, mitigating, attending and participating in the rehabilitation and reconstruction of damage arising from disasters. Responsible for coordinating with public and private institutions, national and international organizations, civil society at various regional and sectoral levels, on matters relating to disaster risk management as a strategy contributing to sustainable development in Guatemala. Website: www.conred.gob.gt |
| INSIVUMEH | Instituto Nacional de Sismología, Vulcanología, Meteorología e Hidrología *(National Institute for Seismology, Volcanology, Meteorology and Hydrology)* | Established in 1976 as a scientific agency of the Guatemalan government. Responsible for the monitoring of hazards across areas of seismology, volcanology, meteorology and hydrology. Tasked with communicating this information to other government agencies, to inform decision-making. Website: www.insivumeh.gob.gt |




**Table 3**. **Examples of five diverse evidence types that might indicate the relevance of a given multi-hazard interaction.**

| Evidence Types | Examples |
|---|---|
| 1. Publications and Reports | Public and confidential government, technical, private sector and/or civil society reports |
| | Peer-reviewed and other research publications |
| | Maps and archive documents |
| | Student projects (e.g., dissertations, theses) |
| | Books |
| | Diaries |
| 2. Social and Other Media | Photographs and video clips (e.g., from print and online newspapers, blogs, websites, tweets, citizen science) |
| | Newspaper articles |
| | Social media posts (e.g., 'Tweets') |
| 3. Field Evidence | Observations from the impact on the built environment (e.g., marks on vertical services to indicate flooding occurred, or the minimum extent flood water reached) |
| | Geological mapping and any field identification of evidence of the hazard occurring (e.g., flood deposits) |
| 4. Stakeholder Engagement | Interviews with the public, hazard professionals, and civil protection officials |
| | Focus Groups |
| | Workshops |
| 5. Miscellaneous | Insurance records |
| | Instrumental records and associated notes |
| | Emergency call out and incident records from emergency services |
| | Remote sensing images |



**Table 4. Five components of a regional interaction framework.** A description of the evidence used in this paper to address and compile the five components of a regional interaction framework.

| Component of Regional Interaction Framework | Relevant Sections | Additional Information and Literature |
|---|---|---|
| Visualisation framework | N/A | Visualisations presented in Gill and Malamud (2014, 2016, 2017), integrated with the conclusions of Gill (2016) to enhance these frameworks. |
| Inclusion and classification of natural hazard types | **Sections 2.2** to **2.6** | Classification of 21 natural hazards (Gill and Malamud, 2014). |
| Population of framework with relevant natural hazard interactions | **Sections 2.2** to **2.6** | Matrix of globally possible interactions (Gill and Malamud, 2014). |
| Examples of networks of hazard interactions | **Sections 2.3** to **2.6** | Visualisation approaches presented in Gill and Malamud (2016). |
| Anthropogenic processes | **Sections 2.2** to **2.6** | Classification of 18 anthropogenic processes (Gill and Malamud, 2017). |



**Table 5. Information bulletin keywords and number of keyword search results.** Six keywords searched for in the information bulletins (English form and abbreviated Spanish verb base), and the number of results generated by each word. Multiple results could be identified in one bulletin.

| English Form | Abbreviated Spanish Verb Base *(used in the keyword search)* | Number of Keyword Search Results *(in the 267 bulletins)* |
|---|---|---|
| Triggering | Desenca… | 0 |
| Provoking | Provoc… | 26 |
| Generating | Genera… | 58 |
| Causing | Caus… | 22 |
| Producing | Produ… | 37 |
| Catalysing | Catál… | 0 |





**Table 6. Case study locations.** Four case study locations and examples of hazard interactions relevant to these locations.

| Details of Field Visit | | Details of Interaction Event | |
|---|---|---|---|
| **Location** | **Date of Field Visit** | **Summary of Interactions** | **Date** *(where appropriate)* |
| Lake Atitlan (San Pedro La Laguna) | 19–29 Jan 2014 | Tropical storm → landslide | 29–30 May 2010 |
| | | Landslide → flooding | Slow, continuous process |
| | | Rainfall → flooding | Slow, continuous process |
| Fuego | 08–12 Feb 2014 | Tephra + rain → lahar → flood | Frequent |
| Lake Atitlan (Tolimán and Panabaj) | 13–15 Feb 2014 | Hurricane → landslide | 5 October 2005 |
| | | Hurricane → landslide → tsunami → flood | October 2005 |
| Santiaguito | 16–19 Feb 2014 | Tephra + rain → lahar → flood | Frequent |



**Table 7. Consideration of Challenges 1–6 (identified in Gill, 2016) with respect to Guatemala.** A description of how Challenges 1–6 are addressed in this regional interaction framework, using stakeholder comments discussed in **Sections 2.5** (interviews) and **Section 2.6** (workshop results) to inform this process.

| Challenge | Relevance in Context of Guatemalan Case Study |
|---|---|
| Spatial Extent | Interview evidence suggested that national and sub-national spatial extents were suitable for regional interaction frameworks. The Southern Highlands of Guatemala, identified in **Figure 7**, includes large population centres and critical infrastructure. We therefore produce regional interaction frameworks for Guatemala (using political boundaries) and the Southern Highlands of Guatemala (using non-political boundaries. For both scales, we consider hazards and interactions that cut across the determined boundaries. |
| Temporal Extent | Interview evidence suggested that regional interaction frameworks be developed for both preparation (before a primary event) and response (immediate aftermath of a primary event). Not all of the natural hazards and interactions will be relevant at any given time. The temporal relevance of interactions may change given a changing set of anthropogenic processes relevant to this region. The temporal relevance of interactions may also change in response to natural and human driven climate change. The frameworks should be viewed as being dynamic, and regularly reviewed and updated to remain relevant. |
| Likelihood-Magnitude Relationships | Interview evidence suggested a desire for additional information on likelihood-magnitude relationships of interactions. This could be done through an expert elicitation method once a completed interaction framework is prepared. Interaction matrices published in this paper can be taken and additional layers of complexity added, according to user requirements. This could include information on likelihood-magnitude relationships or other parameters of interest (e.g., mitigation approaches). |
| Selection and Classification of Hazards | Interview evidence suggested an expanded natural hazards classification would improve understanding and communication of potential hazard interactions. We therefore develop an expanded classification of natural hazards in **Section 3.2**. The review of a broad range of evidence types allows the identification of multiple relevant hazards, seeking to be as comprehensive as possible rather than focusing on specific natural hazard groups. 17 of 21 interview participants (**Section 2.5**) noted anthropogenic processes to be important for consideration, and we discuss these in **Section 3.5**. |
| Identifying Relevant Hazard Interactions | Workshop evidence indicated different stakeholder opinions on the relevance of specific hazard interactions in Guatemala. The use of multiple evidence types can help to populate regional interaction frameworks in a systematic manner. |
| Visualisation Style and User Communities | Interview evidence suggested that a matrix visualisation format would be suitable for hazard and civil protection professionals, our indented user group. We prepare frameworks in English, but these can subsequently be translated into Spanish. Explanations of vocabulary can accompany interaction visualisations. |





**Table 8. Detailed classification of six hazard groups, 19 natural hazard types, and 37 hazard sub-types relevant to Guatemala.**
An outline of a possible hazard classification scheme relevant to Guatemala. Evidence (from **Section 2**) is used to justify the inclusion of each hazard sub-type, and noted in the table, with references from international literature.

| Hazard Group | Hazard Type | Hazard Sub-Type | Evidence A = International Literature B = Civil Protection Bulletins C = Field Observations D = Stakeholder Interviews E = Workshop (≥50% people) | | | | | References (International Literature) |
|---|---|---|---|---|---|---|---|---|
| Geophysical | Earthquake (**EQ**) | Ground Shaking/Rupture | A | | C | D | E | Lindholm *et al.* (2007) |
| | | Liquefaction | A | | | D | | Seed *et al.* (1981); Porfido *et al.* (2014) |
| | Tsunami (**TS**) | Marine Tsunami | A | | | D | E | Fernández and Ortiz (2007) |
| | | Freshwater Tsunami | A | | C | D | E | Siebert *et al.* (2006); Luna (2007) |
| | Volcanic Activity/ Eruption (**VO**) | Subterranean Magma Movement | A | | | D | E | |
| | | Volcanic Explosions (Vertical/Lateral) | A | B | C | D | E | |
| | | Volcanic Gas/Aerosol Emission | A | | | | E | Alvarado *et al.* (2007); Global Volcanism Program (2013); Brown *et al.* (2015) |
| | | Volcanic Ash/Tephra Ejection | A | B | C | D | E | |
| | | Pyroclastic Density Currents | A | B | C | D | E | |
| | | Lava Flows | A | | C | D | E | |
| | Landslide (**LA**) | Submarine Landslide | A | | | | | Von Huene *et al.* (2004); Tappin (2010) |
| | | Subaerial Rockfall | A | B | C | D | E | Rodríguez (2007) |
| | | Subaerial Rotational/Translational Landslide | A | B | C | D | E | Bommer and Rodríguez (2002); Rodríguez (2007) |
| | | Subaerial Debris Flow | A | B | C | D | E | Bucknam *et al.* (2001); Rodríguez (2007); Luna (2007) |
| | | Subaerial Lahar | A | B | C | D | E | Bucknam *et al.* (2001); Harris *et al.* (2006) |
| Hydrological | Flood (**FL**) | Pluvial Flood | A | B | | D | E | Claxton (1986); Stewart and Cangialosi (2012) |
| | | Fluvial Flood | A | B | | D | E | Schuster *et al.* (2001); Harris *et al.* (2006); Soto *et al.* (2015) |
| | | Coastal Flood | A | | | D | E | Cahoon and Hensel (2002) |
| | | Lakeside Flood | A | | C | D | E | Luna (2007) |
| | Drought (**DR**) | Drought | A | | | D | E | Claxton (1986); Hodell *et al.* (2001); Moreno (2006) |
| Shallow Earth Processes (adapted from Hunt, 2005) | Regional Subsidence (**RS**) | Tectonic Subsidence | | | | D | | |
| | Ground Collapse (**GC**) | Karst/Evaporite Collapse | A | | | D | | Cooper and Calow (1998); Kueny and Day (2002) |
| | | Piping Collapse | A | B | | D | E | Stewart (2011); Satarugsa (2011); Hermosilla (2012) |
| | Soil (Local) Subsidence (**SS**) | Soil Shrinkage | A | | | D | E | MAGA/PEDN (2002a) |
| | | Consolidation/Settlement | A | | | | E | Ebmeier *et al.* (2012); Porfido *et al.* (2015) |
| | Ground Heave (**GH**) | Volcanic Inflation/Uplift | A | | C | D | | Johnson *et al.* (2008); Johnson and Lees (2010) |
| | | Soil Expansion (Swelling) | A | | | D | E | MAGA/PEDN (2002a) |
| Atmospheric | Storm (**ST**) | Heavy Rain | A | B | | D | E | MAGA/PEDN (2002b); World Bank (2016) |
| | | Tropical Storm/Hurricane | A | B | | D | E | Pielke Jr *et al.* (2003); Stewart and Cangialosi (2012) |





| Hazard Group | Hazard Type | Hazard Sub-Type | Evidence<br>A = International Literature<br>B = Civil Protection Bulletins<br>C = Field Observations<br>D = Stakeholder Interviews<br>E = Workshop (≥50% people) | | | | References<br>*(International Literature)* |
|---|---|---|---|---|---|---|---|
| | Tornado (**TO**) | Tornado | A | | D | | DesInventar (2016) |
| | Hailstorm (**HA**) | Hailstorm | A | | D | | DesInventar (2016) |
| | Lightning (**LN**) | Lightning | A B | | D | E | NASA (2006); DesInventar (2016) |
| | Extreme Temperature (Heat) (**ET (H)**) | Heatwave | A | | D | E | LAHT (2014) |
| | Extreme Temperature (Cold) (**ET (C)**) | Coldwave/Frost | A | | D | | MAGA (2002); DesInventar (2016) |
| Biophysical | Wildfire (**WF**) | Wildfire | A | C | D | E | Charvériat (2000); IFFN (2002); DesInventar (2016) |
| Space | Geomagnetic Storms (**GS**) | Geomagnetic Storms | | | | | *No location specific evidence, however these are globally relevant natural hazards, and therefore may affect Guatemala.* |
| | Impact Events (**IM**) | Impact Events | | | | | |



**Table 9. Spatial distribution of 37 natural hazard sub-types in Guatemala.** A synthesis table to characterise which regions in Guatemala are susceptible to each of the 37 natural hazard sub-types. Selected regions are (1) low relief northern plateaus, (2) Central Highlands, with deep valleys, (3) Southern Highlands, and (4) Pacific coastal plains.

| Hazard Group | Hazard Type | Hazard Sub-Type | Regions [1,2,3,4] | Evidence A = International Literature B = Civil Protection Bulletins C = Field Observations D = Stakeholder Interviews E = Workshop *(≥50% people)* | | | | |
|---|---|---|---|---|---|---|---|---|
| Geophysical | Earthquake (**EQ**) | Ground Shaking/Rupture | 1, 2, 3, 4 | A | | C | D | E |
| | | Liquefaction | 1, 2, 3, 4 | A | | | D | |
| | Tsunami (**TS**) | Marine Tsunami | 2, 4 | A | | | D | E |
| | | Freshwater Tsunami | 1, 2, 3 | A | | C | D | E |
| | Volcanic Activity/ Eruption (**VO**) | Subterranean Magma Movement | 3 | A | | | D | E |
| | | Volcanic Explosions (Vertical/Lateral) | 3 | A | B | C | D | E |
| | | Volcanic Gas/Aerosol Emission | 3 | A | | | | E |
| | | Volcanic Ash/Tephra Ejection | 1, 2, 3, 4 | A | B | C | D | E |
| | | Pyroclastic Density Currents | 3 | A | B | C | D | E |
| | | Lava Flows | 3 | A | | C | D | E |
| | Landslide (**LA**) | Submarine Landslide | 2, 4 | A | | | | |
| | | Subaerial Rockfall | 1, 2, 3, 4 | A | B | C | D | E |
| | | Subaerial Rotational and Translational Landslide | 1, 2, 3, 4 | A | B | C | D | E |
| | | Subaerial Debris Flow | 1, 2, 3, 4 | A | B | C | D | E |
| | | Subaerial Lahar | 3 | A | B | C | D | E |
| Hydrological | Flood (**FL**) | Pluvial Flood | 1, 2, 3, 4 | A | B | | D | E |
| | | Fluvial Flood | 1, 2, 3, 4 | A | B | C | D | E |
| | | Coastal Flood | 2, 4 | A | | | D | E |
| | | Lakeside Flood | 1, 2, 3 | A | | C | D | E |
| | Drought (**DR**) | Drought | 1, 2, 3, 4 | A | | | D | E |
| Shallow Earth Processes (adapted from Hunt, 2005) | Regional Subsidence (**RS**) | Tectonic Subsidence | 1, 2, 3, 4 | | | | D | |
| | Ground Collapse (**GC**) | Karst/Evaporite Collapse | 1 | A | | | D | |
| | | Piping Collapse | 3 | A | B | | D | E |
| | Soil (Local) Subsidence (**SS**) | Soil Shrinkage | 1, 4 | A | | | D | E |
| | | Consolidation/ Settlement | 1, 2, 3, 4 | A | | | | E |
| | Ground Heave (**GH**) | Volcanic Inflation/Uplift | 3 | A | | C | D | |
| | | Soil Expansion (Swelling) | 1, 2, 3, 4 | A | | | D | E |
| Atmospheric | Storm (**ST**) | Heavy Rain | 1, 2, 3, 4 | A | | B | D | E |
| | | Tropical Storm/Hurricane | 1, 2, 3, 4 | A | B | | D | E |
| | Tornado (**TO**) | Tornado | 1, 2, 3, 4 | A | | | D | |
| | Hailstorm (**HA**) | Hailstorm | 1, 2, 3, 4 | A | | | D | |
| | Lightning (**LN**) | Lightning | 1, 2, 3, 4 | A | B | | D | E |
| | Extreme Temperature (Heat) (**ET (H)**) | Heatwave | 1, 2, 3, 4 | A | | | D | E |
| | Extreme Temperature (Cold) (**ET (C)**) | Coldwave/Frost | 1, 2, 3, 4 | A | | | D | |
| Biophysical | Wildfire (**WF**) | Wildfire | 1, 2, 3, 4 | A | | C | D | E |
| Space | Geomagnetic Storms (**GS**) | Geomagnetic Storms | 1, 2, 3, 4 | | | | | |
| | Impact Events (**IM**) | Impact Events | 1, 2, 3, 4 | | | | | |





**Table 10. Four examples of networks of interactions, extracted from the CONRED civil protection bulletins.** Each example (1–4) is characterised by bulletin number, date, location, and event descriptions.

| Example | Bulletin | | Location | Event Description | Narrative Summary |
|---|---|---|---|---|---|
| | # | Date | | | |
| 1. Storm Alex | 902 | 29-Jun-10 | South West Guatemala | Storm Alex causes floods, landslides/ mudslides. | Heavy rain triggers multiple floods and landslides, with rain also saturating soils and increasing the likelihood of floods in the future. The spatial locations suggest that flooding affected a wide area. |
| | 915 | 02-Jul-10 | Atlantic Coast | Rain associated with hurricane causes flooding. | |
| | 916 | | *General* | Storm Alex causes soils to be saturated and increases likelihood of flooding. | |
| 2. Mixco, Zone 6, Guatemala City | 1062 | 23-Aug-10 | Mixco (Zone 6), Guatemala City | The collapse of a hillside into a river caused damage, with dredging of the river required. | Rain triggers a landslide. This landslide enters a river, which subsequently needs dredging. Landslide therefore either blocked the river and caused flooding or increased the likelihood of flooding. |
| 3. Quetzaltenango Department | 1126 | 09-Sep-10 | Quetzaltenango, Chimaltenango, Alta Verapaz | Rains produced floods, landslides/ mudslides. | Heavy rain in Quetzaltenango and other Departments triggers floods, landslides and lahars. Lahars (requiring ash/tephra deposition) caused flooding of the Samalá river. |
| | 1129 | | San Sebastian, Retalhuleu, Santiaguito | Santiaguito volcano lahars caused flooding of the Samalá river, causing damage to bridges. | |
| 4. Storm Matthew | 1174 | 23-Sep-10 | *General* | Monitoring of rivers during Storm Matthew as it could provoke damage | A warning was issued that Storm Matthew could trigger damage, and was associated with flash floods, landslides and mudslides in Nicaragua and Honduras. On 25 September 2015, Tropical Storm Matthew impacts Guatemala directly, causing river levels to rise and saturate soils, with a warning that flooding may occur. The next bulletins reported flooding, an increased likelihood of landslides, and lightning. |
| | 1175 | 24-Sep-10 | Nicaragua, Honduras | Storm winds and rainfall, cause flash floods, landslides and mudslides | |
| | 1183 | 25-Sep-10 | *General* | Tropical Storm Matthew produces heavy rains which causes rivers to rise. Rains cause soil saturation, expected that rivers will exceed water levels and flooding occur. | |
| | 1184 | | Motagua River, Morales, Izabal | Tropical Storm Matthew causes heavy rains and Motagua river to increase in volume. Overflow of Motagua river caused a flood. | |
| | 1185 | | *General* | Saturated soils could cause landslides or mudslides. | |
| | 1186 | | *General* | Tropical Storm Matthew causes heavy rains, rising tides and floods. | |
| | 1199 | | Centre and South Guatemala | Low pressure system generates clouds, showers and lightning. | |

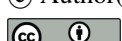



**Table 11. Relevant anthropogenic process types in Guatemala.** A description of the four evidence types A–E, together with additional references, used to identify 17 anthropogenic process types as being spatially relevant in Guatemala.

| Anthropogenic Process Type | Evidence A = International Literature; B = Civil Protection Bulletins; C = Field Observations; D = Stakeholder Interviews; E = Workshop (*anthropogenic processes not discussed*); * (Reference) = Additional citations, beyond A–E. | | | |
|---|---|---|---|---|
| Groundwater Abstraction | | | D | |
| Oil/Gas Extraction | | | | * (OEC, 2016) |
| Subsurface Infrastructure Construction | A | | D | |
| Subsurface Mining | | | | * (OEC, 2016) |
| Material (Fluid) Injection | | | | * (USGeothermal, 2016) |
| Vegetation Removal | A | C | D | |
| Agricultural Practice Change | | C | D | |
| Urbanisation | | C | D | |
| Infrastructure Construction (Unloading) | | | D | |
| Quarrying/Surface Mining (Unloading) | | | | * (OEC, 2016) |
| Infrastructure (Loading) | | C | D | |
| Infilled (Made) Ground | A | | | |
| Reservoir and Dam Construction | A | | D | * (Salini Impregilo, 2014) |
| Drainage and Dewatering | A B | | D | |
| Water Addition | A B | | D | |
| Chemical Explosion | | | | *Inferred relevant* |
| Fire | | | D | |





**Table 12. Calculation of Matthews' Correlation Coefficient (*MCC*) to assess agreement between the collective knowledge of 16 workshop participants (Figure 2) and national interaction framework (Figure 5).** Three different thresholds, each relating to the number of workshop participants (out of 16) identifying a particular interaction, are used to determine collective knowledge of hazard interactions. The number of 'agreements' and 'disagreements' between the workshop participants' response and national interaction framework (see column headers for descriptions) is shown. For each row, the sum of True Positives (TP) and False Negatives (FN) is 50, and the sum of True Negatives (TN) and False Positives (FP) is 392. *MCC* values are determined using **Equation 1**. An *MCC* score of +1.0 means complete agreement; an *MCC* score of −1.0 means complete disagreement.

| Workshop Participants Identifying an Interaction (n = 16) | # Interactions Identified by ≥ x participants (TP + FP) | AGREEMENT [Participants' Collective Framework and National Interaction Framework Agree] | | DISAGREEMENT [Participants' Collective Framework and National Interaction Framework Do Not Agree] | | Matthews' Correlation Coefficient (Equation 1) |
|---|---|---|---|---|---|---|
| | | Interaction Occurs in Both Frameworks | Interaction Does Not Occur in Either Framework | Interaction Occurs in National Framework but not Participants' Collective Framework | Interaction Occurs in Participants' Collective Framework but not National Framework | |
| | | True Positives (TP) | True Negatives (TN) | False Negatives (FN) | False Positives (FP) | |
| ≥ 1 | 86 | 25 | 330 | 25 | 61 | 0.28 |
| ≥ 3 | 32 | 22 | 381 | 28 | 10 | 0.51 |
| ≥ 5 | 19 | 16 | 388 | 34 | 3 | 0.49 |

