# Peer review of "Regional Interaction Frameworks to Support Multi-Hazard Approaches to Disaster Risk Reduction (With an Application to Guatemala)"

_Natural Hazards and Earth System Sciences, 2018_

## Referee Comment (RC1) · Huggel (Referee) · 28 Jan 2019

The research on multi-hazards has increased in recent years, recognizing their importance for generating and exacerbating hazards. Several frameworks and approaches have been developed and applied, and this paper nicely considers them here. Multi- and cascading hazards are probably of particular relevance to developing countries, such as in Central America and Guatemala. I basically like the approach taken here to draw on diverse sources of information and also include stakeholders of the country. The process is transparently described, yet not in a very clear and coherent way. This brings me to my first main point: the paper, and in particular sections 2 and 3, are

quite hard to follow and often somewhat confusing (e.g. the different frameworks and matrices, regional, national, sub-national). As detailed below I think there is potential to shorten and streamline and simplify the text. The methods and results are merged in section 3. The authors may consider separating methods and results in two sections. I'm aware that there may be an issue because the framework, and thus the methods, are somehow representing the results. I recommend to clarify and point the reader more specifically to this issue (whatever the authors chose eventually as an approach to this problem). There is a large number of figures and an excessive number of tables in the paper. I think tables 2, 4, 5, 6 could be removed, and I have some question marks for tables 10 and 11 (see below). The other main point, maybe more fundamental, is the following one: I'm wondering what do we finally learn from this study? Although I appreciate and recognize the important efforts made to collect information from a large set of diverse sources and interacting with stakeholders, the result is a relatively simple matrix which I consider to be a bit thin for a journal paper. This point becomes especially acute if you consider that this same matrix and framework was already developed and presented in the previous Gill and Malamud 2016 and 2017 papers. Do the authors think it is justifiable to yet publish another paper which presents basically the same result with (in my opinion) only little additional substance by applying it to Guatemala? The substance may actually be there, i.e. in the many sources studied, but it is currently hardly in the paper. The authors may therefore reconsider how they present what they have researched (cf my comments below). For instance, I hoped to find more quantitative information regarding the physical processes, e.g. how often do such interactions occur? I'm aware that with the approach taken providing quantitative information related to the physical processes may not be so evident but I would like to encourage the authors to think about it. Most of, but not all, the interactions are quite obvious and well known, such as storms generating floods and landslides. In fact, most of the paper, including the matrices, focus on two interacting processes but the most interesting aspect I found were the cascading hazards (more than 2 processes involved) but unfortunately they receive only little space. Is it possible to extend this

issue, beyond the two case studies (and possibly at the expense of sections 2 and 3 which could be shortened)? Finally, my impression was that some more reflection is needed by the authors. The paper sometimes has more project report character, leaving the reader with a feeling that the authors were short of time. One would like to see more synthesis and less details that are often not particularly relevant. I suggest that the authors take sufficient time to reflect on the objectives and the research questions (both not mentioned in the text) and what can be learned; also how this study contributes to scientific progress. Especially the last point is not evident for me and is not addressed in the paper either. Overall, I'm not sure whether the authors will be able to revise the paper in a round of major revisions in a way that is in my opinion needed, or whether they would rather like to take their time to re-submit it at a later stage.

Specific comments:

Introduction: I think this section could benefit from more text on the processes. The complexities of interacting hazard processes seem to find little attention.

Section 3.3.1: this is an example of a section which is quite confusing to read. The six points made towards the end are not really clear and are they needed?

Section 3.4: as mentioned, I found this the most interesting (and probably novel) section but it is not strongly developed. Is a more quantitative analysis possible?

Page 17, lines 1-3: another option could be to work with / engage researchers with appropriate level of Spanish language.

Page 17, lines 4-12: a very important point in my experiences working in such socio-cultural environments. It applies in particular if risks are considered.

Page 17, lines 25-28: what are the implications of this points?

Discussion section: has interesting and important elements for people working in similar environments. As indicated above, I would like to see more reflection on how this paper advances research on multi-hazards.

Figure 1: not sure this Figure is needed. Considered that the hazard codes need to be explained which is only done is subsequent figures.

Figure 5: many place words are not particularly well readable.

Figure 9: I was wondering whether the color code and the symbols are really used in this figure?

Table 8: I appreciate the level of detail in this table. But it was not clear to me how the hazard sub-types are then used? It is rather just a list which has a value in its own but no further relevance for the paper?

Table 10: I'm not sure how well this table informs us. I found it rather confusing. We see the different bulletin reports which are not necessarily in a logical order (reflecting some issue there) and then the narrative summary. What is really the purpose of this table?

Table 11: I'm not convinced that this table and information needs to be part of the paper (and then probably the respective section as well). Please re-consider.

Christian Huggel, University of Zurich

---

## Referee Comment (RC2) · Kirsten v. Elverfeldt (Referee) · 19 Feb 2019

Summary The paper deals with the development of regional interaction frameworks for Guatemala by utilizing literature reviews, field observations, interviews, and workshops. With the information thus gathered, a classification scheme of natural hazards is determined. Matrices were used to further determine hazard interactions, with a strong focus on the interaction (triggering or increasing the possibility) of two hazards.

Review summary

1. Does the paper address relevant scientific questions within the scope of NHESS?

[Figure]

The paper falls into the subject areas of NHESS. It might fit the scope to understand the behaviour of hazardous natural events.

2. Does the paper present novel concepts, ideas, tools, or data? This remains rather unclear since the authors do not explicitly state the aims, research questions, hypotheses, and novelties of the paper.

3. Are the scientific methods and assumptions valid and clearly outlined? Assumptions are not made explicit. Methods are valid and transparently explained, but explanations would need streamlining and re-structuring.

4. Are the results sufficient to support the interpretations and conclusions? Yes, though the novelty of the results needs to be stressed. I have the feeling that there could be more to the paper than the authors actually delivered. It is difficult to review this paper because the authors leave it to the reader to "read between the lines" and to draw conclusions by herself/himself. In a nutshell, it remains somewhat unclear what we gain by the paper.

5. Do the authors give proper credit to related work and clearly indicate their own new/original contribution? Yes.

6. Does the title clearly reflect the contents of the paper? Yes.

7. Does the abstract provide a concise and complete summary? Yes. However, in the abstract research questions, hypotheses, aims,. . . are missing (as they are in the text).

8. Is the overall presentation well-structured and clear? No. Needs to be improved.

9. Is the language fluent and precise? Yes.

10. Are the number and quality of references appropriate? Yes.

Detailed review - specific comments

Main points

Line 3: The authors start very abruptly with the topic of regional interaction frame-works, without really framing their topic. They present the term "regional interaction framework" right at the beginning, whilst the definition of the term only comes one paragraph later.

Line 4: It remains unclear in how far your approach is interdisciplinary. Even more so, it remains unclear what "the approach" is that is being applied. I suggest that at least (!) a citation of the previous Gill and Malamud papers on this subject should be given here; it'd be even better to continue (after framing your topic) with briefly explaining what your approach is. In general, the writing style of section 1.1. is rather additive than providing an argument for why the study is relevant or in what context it is to be understood. The aim of the paper remains unclear as well as hypotheses, assumptions, and research questions.

Page 3:

Line 13: Is Table 2 necessary? Please consider deleting the table.

Line 20: Here, you distinguish between hazard interactions on the one hand and net-works of interactions (cascades) on the other hand, whilst on page 2 you summarized all interrelated effects (including cascades) under the umbrella of the term hazard in-teractions. Please consider handling this coherently. To me, section 1 is rather over-structured. For example, section 1.3 consists of only three sentences. I'd suggest to re- and de-structure the section, including a better framing of the topic and to be less descriptive and additive, and to put up an argumentation.

Line 12: Suggestion to delete table 4.

Line 30f: I'd also suggest deleting table 5.

Line 6: Suggestion to delete Table 6

Line 16: Here, and at quite a few instances before and afterwards, you refer to later sections in the paper. This makes reading rather difficult and raises the question whether the paper could be structured more coherently. If you discussed the workshops in section 2.6, why do you discuss their limitations so much later in the paper? As a rule of thumb references to content delivered later in a paper should be avoided.

Page 9-10

Lines 15ff: In the paper, comparatively long sections are dedicated to referencing to previous or later content. Suggestion to shorten and re-structure the paper.

Line 8ff: this explanation of what is required for regional interaction frameworks comes at a rather late stage. Since you mention regional interaction frameworks so often on previous pages, I'd suggest to bring together issues that belong together. This would also decrease the amount of references to previous and later sections in the paper. The paper in its current stage is rather difficult to read and readers might easily lose track of what is the intention of the paper or a section in its own.

Line 19: this has been mentioned before (on page 2)

Line 4: In table 8, A-E are named differently from what was proposed in the text.

Line 15: Figure 4: I am not sure that it is useful to have the same figure as in figure 3 repeated only to deliver the information of how many evidence sources were used.

I think it is enough to deliver this information via text only (the number of figures and tables is really high for this paper, and not all of them seem to be necessary).

Line 14: Figure 5 – again, I'd expect to get this information much earlier, e.g. in section 1.2. In table 9, evidence categories A-E differ again from text

Line 24: Figure 6 text is too small, rather impossible to read; is it upside down?

Line 18ff: I cannot quite see the difference between example 1 and 4 (Table 10)? It would also be helpful if you explained what you mean by "linear event", "multi-branch event" etc. This again is some example for how you (superficially) describe rather than explain or argue.

Line 13: In table 11, evidences A-E differ from text

Line 1: It would be useful if you explained and/or detailed the "useful insights" that are generated. I really do like the way you collate information via different methods (literature, interviews, workshops etc.). But I think your paper stops when it gets most interesting, i.e. hazard cascades/networks and anthropogenic impacts on hazard interactions. Furthermore, since you do not explain what you gain aside from a visualisation and collection of (maybe more or less) known hazard interactions, this important aspect remains far too vague. This might also be because the reader doesn't know your aims, hypotheses, and research questions.

Line 10: This is another example that re-structuring the paper is necessary. The limitations and uncertainties should be mentioned where you present the respective method; here, you can then focus on the discussion.

Line 29: I'm confused by the additional information about translators – have you used them? If not, why? If you did, this should be mentioned earlier.

Line 25ff: Plus, if you use a pre-defined hazard scheme without the option to add other hazards and interactions, participants' knowledge might be missed out.

Line 22: Table 9 – colour code and symbol code (legend) to be deleted

Line 17: Why did you set these thresholds and not others? Explanation would be good.

Minor points

Line 11: , and evidenced . . .

Line 15f: to reduce the number of parantheses, I'd suggest to re-write a part of the sentences as follows: (internationally accessible: 93 peer-review and 76 grey literature sources); (locally accessible civil protection bulletins: 267 bulletins from 11 June 2010 to 15 October 2010)

Line 3: , and evidenced . . .

Line 13: Delete "Here, and"

Line 27: Put "and" in italics (two times)

Line 29: Consider rephrasing: . . ."that our approach also supports implementation"

Line 5: , and surface collapses

Line 7: , and cold spells

Line 23: for coherence, I'd suggest to change the heading to "... regional interaction framework"

Line 25: evidences

Line 3: , and media reports (please check for "comma + and" throughout the document).

Line 6: consider rephrasing "an overview of Guatemala's hazard-forming"

Line 15: include is repetitive in the sentence

Line 17: verb missing?

Line 2: helped identifying

Line 7: selected locations

Line 9: Delete "."

Line 2: "." is missing

Please also note the supplement to this comment:
https://www.nat-hazards-earth-syst-sci-discuss.net/nhess-2018-363/nhess-2018-363-RC2-supplement.pdf

[Figure]

**Supplement:**

**Review of nhess-2018-363 - Regional Interaction Frameworks to Support Multi-Hazard Approaches to Disaster Risk Reduction (With an Application to Guatemala).**

**Summary**

The paper deals with the development of regional interaction frameworks for Guatemala by utilizing literature reviews, field observations, interviews, and workshops. With the information thus gathered, a classification scheme of natural hazards is determined. Matrices were used to further determine hazard interactions, with a strong focus on the interaction (triggering or increasing the possibility) of two hazards.

**Review summary**

1. Does the paper address relevant scientific questions within the scope of NHESS?
   The paper falls into the subject areas of NHESS. It might fit the scope to understand the behaviour of hazardous natural events.

2. Does the paper present novel concepts, ideas, tools, or data?
   This remains rather unclear since the authors do not explicitly state the aims, research questions, hypotheses, and novelties of the paper.

3. Are the scientific methods and assumptions valid and clearly outlined?
   Assumptions are not made explicit. Methods are valid and transparently explained, but explanations would need streamlining and re-structuring.

4. Are the results sufficient to support the interpretations and conclusions?
   Yes, though the novelty of the results needs to be stressed. I have the feeling that there could be more to the paper than the authors actually delivered. It is difficult to review this paper because the authors leave it to the reader to "read between the lines" and to draw conclusions by herself/himself. In a nutshell, it remains somewhat unclear what we gain by the paper.

5. Do the authors give proper credit to related work and clearly indicate their own new/original contribution?
   Yes.

6. Does the title clearly reflect the contents of the paper?
   Yes.

7. Does the abstract provide a concise and complete summary?
   Yes. However, in the abstract research questions, hypotheses, aims,… are missing (as they are in the text).

8. Is the overall presentation well-structured and clear?
   No. Needs to be improved.

9. Is the language fluent and precise?
   Yes.

10. Are the number and quality of references appropriate?
    Yes.

**Detailed review - specific comments**

**Main points**

*Page 2*

Line 3: The authors start very abruptly with the topic of regional interaction frameworks, without really framing their topic. They present the term "regional interaction framework" right at the beginning, whilst the definition of the term only comes one paragraph later.

Line 4: It remains unclear in how far your approach is interdisciplinary. Even more so, it remains unclear what "the approach" is that is being applied. I suggest that at least (!) a citation of the previous Gill and Malamud papers on this subject should be given here; it'd be even better to continue (after framing your topic) with briefly explaining what your approach is.

In general, the writing style of section 1.1. is rather additive than providing an argument for why the study is relevant or in what context it is to be understood. The aim of the paper remains unclear as well as hypotheses, assumptions, and research questions.

*Page 3:*

Line 13: Is Table 2 necessary? Please consider deleting the table.

Line 20: Here, you distinguish between hazard interactions on the one hand and networks of interactions (cascades) on the other hand, whilst on page 2 you summarized all interrelated effects (including cascades) under the umbrella of the term hazard interactions. Please consider handling this coherently.

To me, section 1 is rather over-structured. For example, section 1.3 consists of only three sentences. I'd suggest to re- and de-structure the section, including a better framing of the topic and to be less descriptive and additive, and to put up an argumentation.

*Page 4*

Line 12: Suggestion to delete table 4.

*Page 5*

Line 30f: I'd also suggest deleting table 5.

*Page 7*

Line 6: Suggestion to delete Table 6

*Page 8*

Line 16: Here, and at quite a few instances before and afterwards, you refer to later sections in the paper. This makes reading rather difficult and raises the question whether the paper could be

structured more coherently. If you discussed the workshops in section 2.6, why do you discuss their limitations so much later in the paper? As a rule of thumb references to content delivered later in a paper should be avoided.

*Page 9*

Line 15ff: I don't find the letters A-E very helpful. If you want to abbreviate, please consider using 'meaningful' abbreviations such as IPR (international publications & reports), LPR, FO etc.

*Page 9-10*

Lines 15ff: In the paper, comparatively long sections are dedicated to referencing to previous or later content. Suggestion to shorten and re-structure the paper.

*Page 10*

Line 8ff: this explanation of what is required for regional interaction frameworks comes at a rather late stage. Since you mention regional interaction frameworks so often on previous pages, I'd suggest to re-structure the paper so that you bring together issues that belong together. This would also decrease the amount of references to previous and later sections in the paper. The paper in its current stage is rather difficult to read and readers might easily lose track of what is the intention of the paper or a section in its own.

Line 19: this has been mentioned before (on page 2)

*Page 12*

Line 4: In table 8, A-E are named differently from what was proposed in the text.

Line 15: Figure 4: I am not sure that it is useful to have the same figure as in figure 3 repeated only to deliver the information of how many evidence sources were used. I think it is enough to deliver this information via text only (the number of figures and tables is really high for this paper, and not all of them seem to be necessary).

*Page 13*

Line 14: Figure 5 – again, I'd expect to get this information much earlier, e.g. in section 1.2. The paper needs to be better structured. In table 9, evidence categories A-E differ again from text

Line 24: Figure 6 text is too small, rather impossible to read; is it upside down?

*Page 14*

Line 18ff: I cannot quite see the difference between example 1 and 4 (Table 10)? It would also be helpful if you explained what you mean by "linear event", "multi-branch event" etc. This again is some example for how you (superficially) describe rather than explain or argue.

*Page 15*

Line 13: In table 11, evidences A-E differ from text

*Page 16*

Line 1: It would be useful if you explained and/or detailed the "useful insights" that are generated. I really do like the way you collate information via different methods (literature, interviews, workshops etc.). But I think your paper stops when it gets most interesting, i.e. hazard cascades/networks and anthropogenic impacts on hazard interactions. Furthermore, since you do

not explain what you gain aside from a visualisation and collection of (maybe more or less) known hazard interactions, this important aspect remains far too vague. This might also be because the reader doesn't know your aims, hypotheses, and research questions.

Line 10: This is another example that re-structuring the paper is necessary. The limitations and uncertainties should be mentioned where you present the respective method; here, you can then focus on the discussion.

Line 29: I'm confused by the additional information about translators – have you used them? If not, why? If you did, this should be mentioned earlier.

*Page 17*

Line 25ff: Plus, if you define a hazard scheme without the option to add other hazards and interactions, participants' knowledge might be missed out.

*Page 18*

Line 22: Table 9 – colour code and symbol code (legend) to be deleted

*Page 19*

Line 17: Why did you set these thresholds and not others? Explanation would be good.

**Minor points**

*Page 1*

Line 11: , and evidenced …

Line 15f: to reduce the number of parantheses, I'd suggest to re-write a part of the sentences as follows: (internationally accessible: 93 peer-review and 76 grey literature sources)
(locally accessible civil protection bulletins: 267 bulletins from 11 June 2010 to 15 October 2010)

*Page 2*

Line 3: , and evidenced …

Line 13: Delete "Here, and"

Line 27: Put "and" in italics (two times)

Line 29: Consider rephrasing: …"that our approach also supports implementation"

*Page 3*

Line 5: , and surface collapses

Line 7: , and cold spells

Line 23: for coherence, I'd suggest to change the heading to "… regional interaction framework"

Line 25: evidences

*Page 4*

Line 3: , and media reports (please check for "comma + and" throughout the document).

Line 6: consider rephrasing "an overview of Guatemala's hazard-forming"

Line 15: change to "include"; include is repetitive in the sentence

Line 17: verb missing?

*Page 7*

Line 2: maybe rather "to proof" than "to evidence"; helped identifying

Line 7: selected locations

*Page 16*

Line 9: Delete "."

*Page 19*

Line 2: "." is missing

---

## Author Comment (AC1) · 29 Mar 2019

**Journal: NHESS**

**Title: Regional Interaction Frameworks to Support Multi-Hazard Approaches to Disaster Risk Reduction (With an Application to Guatemala)**

**Authors: Joel C. Gill, Bruce D. Malamud, Edy Manolo Barillas, and Alex Guerra Noriega**

**MS No.: nhess-2018-363**          **MS Type: Research article**

Dear Editor, Prof. Dr. Christian Huggel (Reviewer 1), Dr. Kirsten v. Elverfeldt (Reviewer 2)

We are grateful to both reviewers for the constructive way in which they have engaged with this manuscript, and for their detailed comments. In the following response, we address each of these comments in turn, setting out the referee comments (RC1, RC2, etc.), and author comments (AC1, AC2, etc.), including a description of how the manuscript will be changed. To help the editor and reviewers, we have included this cover note with a summary of the major changes and additions we are proposing.
* * *
Both reviewers have highlighted the need for more framing of this manuscript's research and ideas, and an enhanced discussion of why the manuscript might be of interest to others outside of Guatemala. We currently have a second manuscript in review (*International Journal of Disaster Risk Science*) which sets the scene for some of what we have presented in this NHESS submission by assessing the key challenges of constructing and populating regional interaction frameworks—in other words considering the much broader and philosophical implications of going from global to regional multi-hazard frameworks. While we considered the benefits of bringing the two manuscripts together into one submission, we decided that there would be too much information for one submission, and thus divided them into two manuscripts:

> **Manuscript A (IJDRS).** Identifies, characterises, and makes recommendations as to how to address the principal challenges of developing hazard interaction frameworks for use in regional settings.

> **Manuscripts B (NHESS).** Presents an interdisciplinary approach to developing comprehensive, systematic and evidenced regional interaction frameworks to support multi-hazard approaches to disaster risk reduction. We apply this approach in Guatemala, developing regional interaction frameworks for national and sub-national (Southern Highlands) spatial extents.

We now recognise that in splitting them we lost some of the broader framing in Paper B (submitted to NHESS), and therefore more framing is needed in the NHESS manuscript to illustrate (i) the current complexities of constructing regional interaction frameworks, and (ii) how the approach we set out in the NHESS manuscript helps to advance this theme.

We believe that our response to reviewers, and the proposed changes (summarised below), will help to address the disconnect between what we have presented in our manuscript to NHESS and the broader multi-hazard literature. We appreciate both reviewers bringing this to our attention and agree with this general sentiment expressed in their reviewer comments. Here is how we broadly propose to modify our NHESS manuscript:

- *Section 1: Introduction*. We will restructure the introduction, to better articulate our research questions and hypotheses, as well as framing our work in the context of the complexities of studying hazard interactions. We will also integrate these into the abstract. We will highlight our three main research questions:

  o *For a defined spatial region, how does one construct and populate a synthesis of all relevant potential natural hazard interactions using blended sources of evidence for past case histories and theoretical future possibilities from that region's characteristics? [We develop, confront and discuss an approach for Guatemala that has broader relevance and applicability].*

o *How do interactions documented in the literature contrast with the knowledge of hazard/civil protection professionals operating in the region?*

o *What are the implications of our regional interaction frameworks for multi-hazard methodologies to support disaster risk reduction, management and response?*

We address these by collating and uniting diverse evidence sources, from multiple disciplines, through a visual database (i.e., a matrix) of potential interactions. We demonstrate an approach that is *comprehensive* (includes a broad array of potential hazards), *systematic* (exploring the potential for interactions in Guatemala between each hazard pairing) and *evidenced* (documenting the evidence for the existence of interactions).

- *Section 2*: *Evidence Used to Inform the Regional Framework.* We will restructure this section so that it makes clear that we are setting out (i) our data (evidence types) and (ii) the methods used to collect and unite this to address our research questions (now more clearly articulated in **Section 1**). We will carefully review if we can further edit **Section 2** to make it more streamlined, moving material to the **Supplementary Material** if necessary.

- *Section 3: Regional Interaction Frameworks (Visualisations).* We will expand **Section 3.4** on networks of hazard interactions (or cascades), to include more examples from the evidence collected, and an expanded discussion of the importance of considering such networks.

- *Section 4: Discussion.* We will move the limitations to **Section 2**, and expand the discussion of how the methods developed in and results of this paper can help to improve both disaster risk reduction practice and advance multi-hazard research. We will characterise the challenges of adding quantitative information to the matrices we present, and outline potential future research directions to move this forward.

- *Figures and Tables:* We have suggested removing Tables 2, 4, 5, 6 and 11 (moving material to the Supplementary Material where necessary), and simplifying Table 10. We will remove Figures 1 and 4, and edit Figures 5, 6, and 9 to make them easier to read, removing unnecessary information.

We thank the reviewers again for their detailed and constructive reviews, and hope we are given the opportunity to implement their suggestions and enhance this manuscript to their satisfaction.

Kind regards,
Joel Gill (on behalf of all authors)

**Reviewer 1: Christian Huggel**

[RC1] The research on multi-hazards has increased in recent years, recognizing their importance for generating and exacerbating hazards. Several frameworks and approaches have been developed and applied, and this paper nicely considers them here. Multi and cascading hazards are probably of particular relevance to developing countries, such as in Central America and Guatemala. I basically like the approach taken here to draw on diverse sources of information and also include stakeholders of the country. The process is transparently described, yet not in a very clear and coherent way. This brings me to my first main point: the paper, and in particular sections 2 and 3, are quite hard to follow and often somewhat confusing (e.g. the different frameworks and matrices, regional, national, sub-national). As detailed below I think there is potential to shorten and streamline and simplify the text. The methods and results are merged in section 3. The authors may consider separating methods and results in two sections. I'm aware that there may be an issue because the framework, and thus the methods, are somehow representing the results. I recommend to clarify and point the reader more specifically to this issue (whatever the authors chose eventually as an approach to this problem). There is a large number of figures and an excessive number of tables in the paper. I think tables 2, 4, 5, 6 could be removed, and I have some question marks for tables 10 and 11 (see below).

[AC1] We recognise that both reviewers have noted the need to streamline the paper and simplify the text. We will restructure **Section 2** so that it makes clear that we are setting out both our data (evidence types) and the methods used to collect and unite this data to address our research questions (which we will better set out in the Introduction). As suggested by the reviewer, we will indicate to the reader that separating the methods from the results is challenging given the matrix output being both part of the method and the results. We will carefully review if we can further edit **Sections 2** and **3** to make them more coherent, moving material to the Supplementary Material if necessary. As outlined further in our responses below, we have proposed removing five tables from the manuscript (either entirely, or moving to Supplementary Material), and simplifying/shortening one other, as well as removing two figures.

[RC 2] The other main point, maybe more fundamental, is the following one: I'm wondering what do we finally learn from this study? Although I appreciate and recognize the important efforts made to collect information from a large set of diverse sources and interacting with stakeholders, the result is a relatively simple matrix which I consider to be a bit thin for a journal paper. This point becomes especially acute if you consider that this same matrix and framework was already developed and presented in the previous Gill and Malamud 2016 and 2017 papers. Do the authors think it is justifiable to yet publish another paper which presents basically the same result with (in my opinion) only little additional substance by applying it to Guatemala? The substance may actually be there, i.e. in the many sources studied, but it is currently hardly in the paper. The authors may therefore reconsider how they present what they have researched (cf my comments below). For instance, I hoped to find more quantitative information regarding the physical processes, e.g. how often do such interactions occur? I'm aware that with the approach taken providing quantitative information related to the physical processes may not be so evident but I would like to encourage the authors to think about it.

[AC2] We believe that this paper builds on the global approaches that we set out in our Gill and Malamud (2014, 2017) papers, and refines/applies these to help characterise potential hazard interactions at national and sub-national spatial scales. While the matrices take the same visual form as the ones in Gill and Malamud (2014), the approach we have used to construct and populate these matrices are significantly enhanced. We would emphasise that what we are presenting in this paper is not just a matrix output, but also a discussion of a process to go from global to regional scales (interdisciplinary, multi-method approach) that enables the development of comprehensive, systematic and evidenced overviews of potential hazard interactions. We are presenting a suite of visualisations that build on our Gill and Malamud (2014 and 2017) papers with a greater range of hazard types, and matrices that are populated using different evidence. In Gill and Malamud (2014), done at a 'global' and high-level scale, we relied on published literature and case studies, whereas in

our NHESS manuscript we have integrated diverse evidence types including interviews, data generating workshops, and interrogation of civil protection bulletins.

We therefore point to the substance of this paper being:

- Development and description of an interdisciplinary, multi-method approach that enables the development of comprehensive, systematic and evidenced overviews of potential hazard interactions at a regional (e.g., national/sub-national) scale. This contrasts with the existing studies of potential interactions, which are generally selective about which hazards they include and do not describe the evidence for including/excluding certain hazards or interactions between hazards.

- Application of this approach in the context of Guatemala to produce a suite of comprehensive and robust frameworks of potential hazard interactions for two spatial scales (national and sub-national), and describes their application to disaster risk reduction (including through initial efforts to embed them into key government agencies in Guatemala). The matrices presented include 21 to 33 hazards (compared to 6 to 11 natural hazards in the surveyed literature examples, which we summarize in **Table 1** of our manuscript).

- Description of an approach for contrasting current individual/collective knowledge with the published regional interaction framework, using Matthews' Correlation Coefficient. The results of this both underpin *why* developing comprehensive and evidenced frameworks of interactions is important (highlighting knowledge gaps), and provides a tool (and we recognize other tools could also be used) to monitor changes in understanding of hazard interactions over time.

We recognise that there are additional layers of information that could be helpfully added to what we have currently presented (e.g., how often each interaction occurs). We do not currently have this information in a level of detail that would be helpful to the reviewer or reader, and it was beyond the scope of our initial research remit (which we acknowledge we need to set out with more clarity in **Section 1**). For each interaction (not hazard), understanding the frequency-magnitude of occurrence and the range of potential impacts would involve significant work and collation of diverse information that we currently do not have (as noted on page 21 of our original manuscript).

We will highlight some of the above directions and intent of our work in our introduction, and also propose enhancing the discussion in **Section 4.3** of quantitative characterisation of hazard interactions, noting current challenges in doing this and how future work could help to enrich this characterisation of potential interactions. We will suggest one approach is using an online wiki-style system where users can click on a cell in the matrix, and upload relevant papers, datasets, or their own assessments of frequency-magnitude to help improve this understanding. We will further, in appropriate places in the manuscript, emphasize what readers outside of the case-study area of Guatemala might learn from our study.

We note that in **Section 4.2** we have set out a quantitative characterisation of hazard and disaster professionals' individual and collective knowledge of hazard interactions – and outlined the significance of this assessment.

[RC3] Most of, but not all, the interactions are quite obvious and well known, such as storms generating floods and landslides. In fact, most of the paper, including the matrices, focus on two interacting processes but the most interesting aspect I found were the cascading hazards (more than 2 processes involved) but

unfortunately they receive only little space. Is it possible to extend this issue, beyond the two case studies (and possibly at the expense of sections 2 and 3 which could be shortened)?

[AC3] (a) The reviewer notes that hazard interactions in the matrices in **Figures 3** and **6** are obvious. We accept that many hazard pairings included in the matrix (e.g., storms triggering floods, earthquakes triggering landslides), and their spatial relevance to Guatemala, are well known. However, we note that this paper is establishing an evidenced framework (expressed as a matrix) of potential interactions. This national scale framework of potential interactions is, we believe, rarely discussed in the current hazard literature, as well as the method for developing a comprehensive and systematic framework. This manuscript has also contrasted the full list of potential interactions in **Figure 3**, with individual and collective knowledge (**Section 4.2**) in the region. The results of this highlight that the spatial relevance of the interactions are not always obvious, and therefore a systematic documentation and visualisation of potential interactions could help. We will make these points clearer in the text. (b) While the matrices focus on how any one single hazard could trigger or increase the probability of another single hazard, it is possible to use these matrices to extract examples of longer cascades. We will make this clearer in the text. We also propose expanding **Section 3.4** on networks of hazard interactions. We will include more examples from the evidence collected, and an expanded discussion of the importance of such networks.

[RC4] Finally, my impression was that some more reflection is needed by the authors. The paper sometimes has more project report character, leaving the reader with a feeling that the authors were short of time. One would like to see more synthesis and less details that are often not particularly relevant. I suggest that the authors take sufficient time to reflect on the objectives and the research questions (both not mentioned in the text) and what can be learned; also how this study contributes to scientific progress. Especially the last point is not evident for me and is not addressed in the paper either. Overall, I'm not sure whether the authors will be able to revise the paper in a round of major revisions in a way that is in my opinion needed, or whether they would rather like to take their time to re submit it at a later stage.

[AC4] We believe that we can address this comment by reviewing both the introduction and the discussion sections, and improving the way in which we frame the work we have done.

- We will clearly articulate our research questions (see the opening cover letter, pp. 1 to 2, and **AC2**) in the introduction, and in doing so help to frame the subsequent sections.
- We will expand the discussion section to outline how this manuscript advances both multi-hazard research, and disaster risk reduction practice.

A key step in understanding risk (Sendai Framework Priority for Action 1) is understanding the hazard landscape of a region (i.e., the relevant single hazards, and how they may interact to generate combinations or cascades of interactions). Currently, regional studies of potential hazard interactions are sparse and none of these set out a replicable and scalable method for systematically doing this. In our paper, we describe and apply an approach that is replicable and can be applied at regional, national and sub-national spatial extents.

In the multi-step, method we present through an application to Guatemala, we include 2 to 3 times the number of natural hazards that other regional studies have included, and (rarely done) comprehensively set out the evidence for these interactions being spatially relevant. We integrate evidence from both natural and social science methodologies to construct a visualisation that – when returned to Guatemala – was shown to provoke cross-hazard and cross-institutional dialogue.

We believe this supports the scientific community to help construct more evidenced and detailed profiles of relevant interactions for diverse user groups, and through these profiles identify specific research and innovation gaps, as well as knowledge exchange and collaboration opportunities. We

will integrate some of these comments, and expand on them, in the discussion section with examples from Guatemala.

*Specific comments:*

[RC5] Introduction: I think this section could benefit from more text on the processes. The complexities of interacting hazard processes seem to find little attention.

[AC5] We will add further detail on the complexities of interacting hazard processes to the introduction.

[RC6] Section 3.3.1: this is an example of a section which is quite confusing to read. The six points made towards the end are not really clear and are they needed?

[AC6] We will improve the clarity of **Section 3.3.1** in order to assist the reader. Points (i) to (vi) in our manuscript are included to describe the contents of **Figure 4**, illustrating that some interactions are well documented in multiple evidence sources whereas others are not well documented. We will review whether this information can be made clearer, and if not we will remove it from the manuscript.

[RC7] Section 3.4: as mentioned, I found this the most interesting (and probably novel) section but it is not strongly developed. Is a more quantitative analysis possible?

[AC7] We will expand **Section 3.4** on networks of hazard interactions. We will include more examples from the evidence collected, and an expanded discussion of the importance of such networks. This will result in (i) profiling of more examples from Guatemala, and (ii) synthesising key implications for disaster risk reduction from such examples. While we do not have the data to apply such a method to any of the scenarios we present, we can point the reader here to existing methods for quantitatively assessing probabilities of specific hazard cascades (e.g., using event scenario trees, such as done by Neri *et al.* (2008, 2013).

[RC8] Page 17, lines 1-3: another option could be to work with / engage researchers with appropriate level of Spanish language.

[AC8] We acknowledge this is one approach that would work, and will include reference to this in the text. This work was initially part of a PhD, and therefore data collection was primarily the responsibility of the lead author.

[RC9] Page 17, lines 4-12: a very important point in my experiences working in such sociocultural environments. It applies in particular if risks are considered.

[AC9] We agree, and felt it was helpful to emphasise in the write up given the importance of the natural science community being more aware of such considerations.

[RC10] Page 17, lines 25-28: what are the implications of this points?

[AC10] We asked workshop participants to describe two different types of interaction: (1) one hazard triggering another hazard, and (2) one hazard increasing the probability of another hazard. If confusion between these two types of interaction existed, it is possible that participants may have inadvertently characterised an interaction as one type when they meant the other (i.e., a specific hazard pairing suggested to be a triggering relationship may actually be means to be communicated as an increased probability relationship). We do not believe the results expressed in **Figures 3** and **6** are unduly influenced by this, given the use of multiple evidence sources to construct them.

[RC11] Discussion section: has interesting and important elements for people working in similar environments. As indicated above, I would like to see more reflection on how this paper advances research on multi-hazards.

[AC11] Our discussion section currently explores how regional interaction frameworks can advance multi-hazard risk reduction. We have focused more on the relationship of our work to practice, rather than research, but can expand the discussion to include more on the latter. We currently explore collective knowledge of hazard interactions in Guatemala, and note that interaction frameworks help to facilitate enhanced cross-institutional dialogue about hazard interactions, their likelihoods and potential impacts. This could help to strengthen collective knowledge of hazard interactions, and the ability of an individual to access this knowledge. We also described the response of hazard and civil protection professionals in Guatemala to our results, and their perspectives on 'next steps'. We also briefly describe in **Section 4.4** how interaction frameworks can help to improve decision making in key agencies engaged in DRR and civil protection. We believe this latter section could be expanded, to further outline how this work advances multi-hazards research. For example, interaction frameworks can guide future research priorities by determining where there is a lack of evidence and/or understanding of certain interactions. See also **AC2, AC4**, **AC36**.

[RC12] Figure 1: not sure this Figure is needed. Considered that the hazard codes need to be explained which is only done is subsequent figures.

[AC12] We will remove Figure 1, and integrate this information into Figure 2.

 [RC13] Figure 5: many place words are not particularly well readable.

[AC13] We will edit the figure, so that it can be larger, with larger text. We believe that place words will be easier to read in a final version where figures are uploaded separately (as an image file), and thus be of a better resolution.

[RC14] Figure 9: I was wondering whether the color code and the symbols are really used in this figure?

[AC14] We agree, and will adjust the legend in this figure to remove this unnecessary information.

[RC15] Table 8: I appreciate the level of detail in this table. But it was not clear to me how the hazard sub-types are then used? It is rather just a list which has a value in its own but no further relevance for the paper?

[AC15] The list presented in **Table 8** was developed from the evidence described in **Section 2**, as a classification of hazards relevant to Guatemala, using categories that many stakeholders in the region would understand. We take our classification and use this as the basis for the analysis in **Figures 6** to **8**. We propose leaving **Table 8** in the manuscript, but expanding **Section 3.2** to outline how this classification is integrated into the rest of the paper.

[RC16] Table 10: I'm not sure how well this table informs us. I found it rather confusing. We see the different bulletin reports which are not necessarily in a logical order (reflecting some issue there) and then the narrative summary. What is really the purpose of this table?

[AC16] The purpose of this table is to demonstrate an approach for identifying relevant, complex cascades that have previously impacted Guatemala. We highlight that while evidence exists for these cascades in a set of civil protection bulletins, they are not outlined in a coherent way but often different strands are included in different bulletins. **Table 10** presents four examples of the cascades that the reviewer highlighted to be particular interesting in **RC3**. We include the bulletin information to connect these examples to the evidence that describes them, but accept that the event description and narrative summary could be combined to make the table more succinct. In **AC3**, we suggested

expanding **Section 3.4** to include a more detailed discussion of cascades. We will therefore expand the text introducing **Table 10** to better articulate its purpose.

[RC17] Table 11: I'm not convinced that this table and information needs to be part of the paper (and then probably the respective section as well). Please re-consider.

> [AC17] We believe that it is important to make reference to anthropogenic processes in this paper, given their ability to trigger and/or catalyse natural hazards. Many stakeholders emphasised the importance of anthropogenic activity in triggering landslides in Guatemala, for example. Reviewer 2 also noted this to be an interesting section (see **RC35**). We have kept the section short and signpost to other literature. We will move **Table 11 t**o the supplementary material.

**Reviewer 2: Kirsten v. Elverfeldt**

[RC18] Summary. The paper deals with the development of regional interaction frameworks for Guatemala by utilizing literature reviews, field observations, interviews, and workshops. With the information thus gathered, a classification scheme of natural hazards is determined. Matrices were used to further determine hazard interactions, with a strong focus on the interaction (triggering or increasing the possibility) of two hazards.

Review summary

1. Does the paper address relevant scientific questions within the scope of NHESS? The paper falls into the subject areas of NHESS. It might fit the scope to understand the behaviour of hazardous natural events.

**2. Does the paper present novel concepts, ideas, tools, or data? This remains rather unclear since the authors do not explicitly state the aims, research questions, hypotheses, and novelties of the paper.**

**3. Are the scientific methods and assumptions valid and clearly outlined? Assumptions are not made explicit. Methods are valid and transparently explained, but explanations would need streamlining and re-structuring.**

**4. Are the results sufficient to support the interpretations and conclusions? Yes, though the novelty of the results needs to be stressed. I have the feeling that there could be more to the paper than the authors actually delivered. It is difficult to review this paper because the authors leave it to the reader to "read between the lines" and to draw conclusions by herself/himself. In a nutshell, it remains somewhat unclear what we gain by the paper.**

5. Do the authors give proper credit to related work and clearly indicate their own new/original contribution? Yes.

6. Does the title clearly reflect the contents of the paper? Yes.

**7. Does the abstract provide a concise and complete summary? Yes. However, in the abstract research questions, hypotheses, aims,: : : are missing (as they are in the text).**

**8. Is the overall presentation well-structured and clear? No. Needs to be improved.**

9. Is the language fluent and precise? Yes.

10. Are the number and quality of references appropriate? Yes.

> [AC18, addressing the comments highlighted in bold above]. The reviewer notes that the manuscript needs some streamlining and restructuring, clearer articulation of assumptions, and added emphasis on the novelty and importance of the results. If given the opportunity to submit a second draft, we will:

- Revise the abstract and introduction (**Section 1**), including making our objectives and research questions clearer, as articulated in the cover letter of this response.
- Revising **Section 2** to streamline our discussion of methods alongside the discussion of data (or evidence) types used.
- Better articulate any assumptions we have made in the introduction, methods and analysis sections.
- Revising the introduction (**Section 1**) and discussion sections (**Section 4**) to make it clearer what the novelty of this paper is, and how this advances disaster risk reduction in multi-hazard contexts (see **AC2** and **AC4**).

[RC19] Page 2, Line 3: The authors start very abruptly with the topic of regional interaction frameworks, without really framing their topic. They present the term "regional interaction framework" right at the beginning, whilst the definition of the term only comes one paragraph later.

[AC19] As noted in **AC5** we will expand the introduction, with more framing of the topic, and its relevance to multi-hazard approaches. We will include a definition of the term *regional interaction framework* at its first use.

[RC20] Page 2, Line 4: It remains unclear in how far your approach is interdisciplinary. Even more so, it remains unclear what "the approach" is that is being applied. I suggest that at least (!) a citation of the previous Gill and Malamud papers on this subject should be given here; it'd be even better to continue (after framing your topic) with briefly explaining what your approach is. In general, the writing style of section 1.1. is rather additive than providing an argument for why the study is relevant or in what context it is to be understood. The aim of the paper remains unclear as well as hypotheses, assumptions, and research questions.

[AC20] In our rewriting of the introduction (see **AC5**, **AC1**8, **AC19**), we will set out the approach that we are following. We will refer to previous literature (including, but not limited to the Gill and Malamud papers that the reviewer highlighted), and outline an explanation of the approach that we will follow. This consists of the uniting and synthesising of diverse evidence types from the natural and social sciences (e.g., field observations, interviews) within a visualisation framework that enables rapid understanding of potential hazard interactions. We argue that the outputs (e.g., **Figures 3** and **6**) are interdisciplinary in that their construction relies on the application of methods from multiple disciplinary fields, and that they synthesis information from diverse specialisms (e.g., hydrology, geology, meteorology, engineering). As noted in **AC18**, we will better articulate our hypotheses, assumptions, and research questions into the introduction.

[RC21] Page 3, Line 13: Is Table 2 necessary? Please consider deleting the table.

[AC21] We will delete this table or move it to the supplementary material.

[RC22] Page 3, Line 20: Here, you distinguish between hazard interactions on the one hand and networks of interactions (cascades) on the other hand, whilst on page 2 you summarized all interrelated effects (including cascades) under the umbrella of the term hazard interactions. Please consider handling this coherently. To me, section 1 is rather overstructured. For example, section 1.3 consists of only three sentences. I'd suggest to re- and de-structure the section, including a better framing of the topic and to be less descriptive and additive, and to put up an argumentation.

[AC22] We will ensure coherency of language throughout the paper to reflect a standard set of terminology. As noted in **AC5** and **AC19** we plan to expand the introduction and include more material to frame this discussion. This will result in **Section 1** being restructured, although our writing style preference is to retain a 'structure of paper' section at the end of the introduction to guide the reader.

[RC23] Page 4, Line 12: Suggestion to delete Table 4.

[AC23] We will delete this table, and include the content within the manuscript text.

[RC24] Page 5, Line 30f: I'd also suggest deleting Table 5.

[AC24] We will delete this table, and include the content within the manuscript text.

[RC25] Page 7, Line 6: Suggestion to delete Table 6

[AC25] We will delete this table, and include the content within the manuscript text.

[RC26] Page 8, Line 16: Here, and at quite a few instances before and afterwards, you refer to later sections in the paper. This makes reading rather difficult and raises the question whether the paper could be structured more coherently. If you discussed the workshops in section 2.6, why do you discuss their limitations so much later in the paper? As a rule of thumb references to content delivered later in a paper should be avoided.

[RC27] Page 9-10, Lines 15ff: In the paper, comparatively long sections are dedicated to referencing to previous or later content. Suggestion to shorten and re-structure the paper.

[RC28] Page 10, Line 8ff: this explanation of what is required for regional interaction frameworks comes at a rather late stage. Since you mention regional interaction frameworks so often on previous pages, I'd suggest to bring together issues that belong together. This would also decrease the amount of references to previous and later sections in the paper. The paper in its current stage is rather difficult to read and readers might easily lose track of what is the intention of the paper or a section in its own.

[AC26] [AC27] [AC28] We will review all references to previous and future content and try to reduce this. We think some of this referencing can be helpful, to signpost to the reader that we are building on something that has come previously, but will work to reduce the prevalence of this type of cross-referencing. We will include more detail on regional interaction frameworks in the introduction (reducing the amount needed in **Section 3.1**) as suggested by the reviewer.

[RC29] Page 10, Line 19: this has been mentioned before (on page 2)

[AC29] We will rephrase this section (page 10) so it better builds on what was presented earlier in the manuscript.

[RC30] Page 12, Line 4: In table 8, A-E are named differently from what was proposed in the text.

[AC30] We will correct any inconsistency between the text and **Table 8**.

[RC31] Page 12, Line 15: Figure 4: I am not sure that it is useful to have the same figure as in figure 3 repeated only to deliver the information of how many evidence sources were used. I think it is enough to deliver this information via text only (the number of figures and tables is really high for this paper, and not all of them seem to be necessary).

[AC31] The purpose of **Figure 4** is to rapidly assess where there could be uncertainty, and future research needed. We do not think this would be easy if the information was presented in the text. We tried to add additional information to **Figure 3**, but this reduces the clarity of this key figure. We propose moving this figure to the Supplementary Material, and referring to it in the figure caption of **Figure 3**.

[RC32] Page 13, Line 14: Figure 5 – again, I'd expect to get this information much earlier, e.g. in section 1.2. In table 9, evidence categories A-E differ again from text

    [AC32] We will move **Figure 5** to the introduction. We will make sure A-E are consistent between **Table 9** and the text.

[RC33] Page 13, Line 24: Figure 6 text is too small, rather impossible to read; is it upside down?

    [AC33] We will increase the text size and reduce the amount of information presented in this figure to increase its clarity.

[RC34] Page 14, Line 18ff: I cannot quite see the difference between example 1 and 4 (Table 10)? It would also be helpful if you explained what you mean by "linear event", "multi-branch event" etc. This again is some example for how you (superficially) describe rather than explain or argue.

    [AC34] We will add further explanation to what we mean by these terms, and enhance the explanations in this section as noted in previous comments (**AC3** and **AC16**). Examples 1 and 4 do have some similarities, and we could remove one from **Table 10** to help make the discussion more succinct. We will also include a simple, visual summary of each example to illustrate the example.

 [RC35] Page 15, Line 13: In table 11, evidences A-E differ from text

    [AC35] We will make sure these are consistent.

[RC36] Page 16, Line 1: It would be useful if you explained and/or detailed the "useful insights" that are generated. I really do like the way you collate information via different methods (literature, interviews, workshops etc.). But I think your paper stops when it gets most interesting, i.e. hazard cascades/networks and anthropogenic impacts on hazard interactions. Furthermore, since you do not explain what you gain aside from a visualisation and collection of (maybe more or less) known hazard interactions, this important aspect remains far too vague. This might also be because the reader doesn't know your aims, hypotheses, and research questions.

    [AC36] We refer to our reply to **AC2** and **AC4**. We would emphasise that what we have gained extends beyond the location-specific visualisation, to a replicable and scalable method that can be applied in other contexts to better understand the hazard landscape. A comprehensive overview of potential hazard interactions allows agencies responsible for hazard monitoring and response to assess if current disaster risk reduction and response strategies, and communication and collaboration mechanisms, can be enhanced to recognise the complexity represented in this paper. As noted in the cover letter, **AC2** and **AC4**, we propose extending **Section 4** to better articulate the significance of what we have developed and how this can be used (and augmented) to improve disaster risk reduction, along with signposting the relevance of our work in the introduction and other selected places in our manuscript. For example, as described in AC11 we will add to **Section 4.4** to describe how interaction frameworks can help to improve decision making in key agencies engaged in DRR and civil protection, such as guiding future research priorities by determining where there is a lack of evidence and/or understanding of certain interactions.

[RC37] Page 16, Line 10: This is another example that re-structuring the paper is necessary. The limitations and uncertainties should be mentioned where you present the respective method; here, you can then focus on the discussion.

    [AC37] Many of the limitations we present cut across multiple evidence types, and therefore the limitations are more succinctly described when presented together. We will, however, move these to end of **Section 2** so that they naturally come after the descriptions of evidence types.

[RC38] Page 16, Line 29: I'm confused by the additional information about translators – have you used them? If not, why? If you did, this should be mentioned earlier.

[AC38] We used a variety of translation methods, and will add a line about this in **Section 2.5** (Stakeholder Engagement: Interviews)

[RC39] Page 17, Line 25ff: Plus, if you use a pre-defined hazard scheme without the option to add other hazards and interactions, participants' knowledge might be missed out.

[AC39] This is true in the context of the workshops where visualisations were provided with the pre-defined hazard scheme included. During workshops, some participants critically examined these 21 natural hazards and clarified that some hazards (e.g., debris flows) were included in the broader descriptions in the scheme (e.g., landslide). We can add this to the discussion of limitations in **Section 4.1**. In interviews, participants were not limited to only discussing the contents of the pre-defined hazard scheme.

[RC40] Page 18, Line 22: Table 9 – colour code and symbol code (legend) to be deleted

[AC40] We will remove this information from **Figure 9**.

[RC41] Page 19, Line 17: Why did you set these thresholds and not others? Explanation would be good.

[AC41] Thresholds of 3 and 5 were selected arbitrarily to demonstrate how this approach could be adjusted to remove those interactions only volunteered by one (or a small number of) professionals, thus acting as a form of quality control. We could have chosen thresholds of 2 or 4, but determined that increments of 1, 3 and 5 would give a spread of results to illustrate the discussion. We do not place great emphasis on the specific threshold in the manuscript, nor try to defend this as being a critical choice. Rather we demonstrate how this approach can help to examine differences between stakeholder perspectives and our national interaction frameworks, and monitor changing understanding and perceptions of natural hazard interactions.

[RC42] Page 1, Line 11: , and evidenced : : :

[RC43] Page 1, Line 15f: to reduce the number of parentheses, I'd suggest to re-write a part of the sentences as follows: (internationally accessible: 93 peer-review and 76 grey literature sources); (locally accessible civil protection bulletins: 267 bulletins from 11 June 2010 to 15 October 2010)

[RC44] Page 2, Line 3: , and evidenced : : :

[RC45] Page 2, Line 13: Delete "Here, and"

[RC46] Page 2, Line 27: Put "and" in italics (two times)

[RC47] Page 2, Line 29: Consider rephrasing: : : :"that our approach also supports implementation"

[RC48] Page 3, Line 5: , and surface collapses

[RC49] Page 3, Line 7: , and cold spells

[RC50] Page 3, Line 23: for coherence, I'd suggest to change the heading to ": : : regional interaction framework"

[RC51] Page 3, Line 25: evidences

[RC52] Page 4, Line 3: , and media reports (please check for "comma + and" throughout the document).

[RC53] Page 4, Line 6: consider rephrasing "an overview of Guatemala's hazard-forming"

[RC54] Page 4, Line 15: include is repetitive in the sentence

[RC55] Page 4, Line 17: verb missing?

[RC56] Page 7, Line 2: helped identifying

[RC57] Page 7, Line 7: selected locations

[RC58] Page 16, Line 9: Delete "."

[RC59] Page 19, Line 2: "." is missing

[AC42–59] We will make these suggested changes and corrections, and thank the reviewer for their detailed review.

---

## Author Response (AR1)

28 June 2019

Dear Professor Fuchs (Editor, NHESS),

Please find in this document an overview of the revisions made to our manuscript NHESS-2018-363 "Regional Interaction Frameworks to Support Multi-Hazard Approaches to Disaster Risk Reduction (With an Application to Guatemala)" by Gill *et al.*.

We thank you for your directions: "*I received the two reports of the referees as well as your comprehensive answer to their comments. I kindly would like to thank you for the numerous and inclusive explanations of what you plan to revise. In general, the topic presented is of considerable interest to the readers of the target journal, and given your reply I kindly would like to ask you to proceed with the revisions of your manuscript. As indicated by both of the referees, the main focus should be on RCs 1-4 and 18-28 according to your numbering in the Author Comment Supplement.  I wish you good success with your work, and I am looking forward to receive a revised version.*"

Our revisions are largely in line with the information we previously prepared and submitted online during the discussion phase in response to the reviewer's comments. Below we include a point by point response, outlining our changes. We have also included a version with all changes tracked, and a clean version of the manuscript with all changes accepted.

Kind regards,

Joel Gill (on behalf of all authors)

**Reply to reviewer comments (NHESS_2018-363)**

We again thank both Christian Huggel and Kirsten v. Elverfeldt as reviewers for their thoughtful and extensive comments. Both reviewers have highlighted the need for more framing of this manuscript's research and ideas, and an enhanced discussion of why the manuscript might be of interest to others outside of Guatemala. We currently have a second manuscript in review (*International Journal of Disaster Risk Science*) which sets the scene for some of what we have presented in this NHESS submission by assessing the key challenges of constructing and populating regional interaction frameworks—in other words considering the much broader and philosophical implications of going from global to regional multi-hazard frameworks. While we considered the benefits of bringing the two manuscripts together into one submission, we decided that there would be too much information for one submission, and thus divided them into two manuscripts:

- **Manuscript A (IJDRS).** Identifies, characterises, and makes recommendations as to how to address the principal challenges of developing hazard interaction frameworks for use in regional settings.
- **Manuscripts B (NHESS).** Presents an interdisciplinary approach to developing comprehensive, systematic and evidenced regional interaction frameworks to support multi-hazard approaches to disaster risk reduction. We apply this approach in Guatemala, developing regional interaction frameworks for national and sub-national (Southern Highlands) spatial extents.

We now recognise that in splitting them we lost some of the broader framing in Manuscript B (submitted to NHESS), and therefore more framing is needed in the NHESS manuscript to illustrate (i) the current complexities of constructing regional interaction frameworks, and (ii) how the approach we set out in the NHESS manuscript helps to advance this theme.

We believe that our response to reviewers, and the changes we have now made (summarised below), these have helped to address the disconnect between what we have presented in our manuscript to NHESS and the broader multi-hazard literature. We appreciate both reviewers bringing this to our attention and agree with this general sentiment expressed in their reviewer comments. Here is how we have modified our NHESS manuscript:

- *Section 1: Introduction*. We have restructured the introduction, to better articulate our research questions, as well as framing our work in the context of the complexities of studying hazard interactions. We have highlighted our three main research questions:

    o *For a defined spatial region, how does one construct and populate a synthesis of all relevant potential natural hazard interactions using blended sources of evidence for past case histories and theoretical future possibilities from that region's characteristics? [We develop, confront and discuss an approach for Guatemala that has broader relevance and applicability].*

    o *How do interactions documented in the literature contrast with the knowledge of hazard/civil protection professionals operating in the region?*

    o *What are the implications of our regional interaction frameworks for multi-hazard methodologies to support disaster risk reduction, management and response?*

    We address these by collating and uniting diverse evidence sources, from multiple disciplines, through a visual database (i.e., a matrix) of potential interactions. We demonstrate an approach that is *comprehensive* (includes a broad array of potential hazards), *systematic* (exploring the potential for interactions in Guatemala between each hazard pairing) and *evidenced* (documenting the evidence for the existence of interactions). We have also updated the abstract.

- *Section 2*: *Evidence Used to Inform the Regional Framework.* We have made clearer in this section that we are setting out (i) our data (evidence types) and (ii) the methods used to collect and unite this to address our research questions (now more clearly articulated in **Section 1**). We have made some edits to **Section 2** to make it more streamlined, moving some material to the **Supplementary Material**.

- *Section 3: Regional Interaction Frameworks (Visualisations).* We have expanded **Section 3.4** on networks of hazard interactions (or cascades), to include more examples from the evidence collected, and an expanded discussion of the importance of considering such networks.

- *Section 4: Discussion.* We have moved the limitations to **Section 2**, and expanded the discussion of how the methods developed in and results of this paper can help to improve both disaster risk reduction practice and advance multi-hazard research. We have characterised the challenges of adding quantitative information to the matrices we present, and outlined potential future research directions to move this forward.

- *Figures and Tables:* We have removed Tables 2, 4, 5, 6 and 11 (moving material to the Supplementary Material where necessary), and simplified Table 10. We have removed Figures 1 and 4, and edited Figures 5, 6, and 9 to make them easier to read, removing unnecessary information.

**Reviewer 1: Christian Huggel**

[RC1] The research on multi-hazards has increased in recent years, recognizing their importance for generating and exacerbating hazards. Several frameworks and approaches have been developed and applied, and this paper nicely considers them here. Multi and cascading hazards are probably of particular relevance to developing countries, such as in Central America and Guatemala. I basically like the approach taken here to draw on diverse sources of information and also include stakeholders of the country. The process is transparently described, yet not in a very clear and coherent way. This brings me to my first main point: the paper, and in particular sections 2 and 3, are quite hard to follow and often somewhat confusing (e.g. the different frameworks and matrices, regional, national, sub-national). As detailed below I think there is potential to shorten and streamline and simplify the text. The methods and results are merged in section 3. The authors may consider separating methods and results in two sections. I'm aware that there may be an issue because the framework, and thus the methods, are somehow representing the results. I recommend to clarify and point the reader more specifically to this issue (whatever the authors chose eventually as an approach to this problem). There is a large number of figures and an excessive number of tables in the paper. I think tables 2, 4, 5, 6 could be removed, and I have some question marks for tables 10 and 11 (see below).

> [AC1] We recognise that both reviewers have noted the need to streamline the paper and simplify the text, and have strived to do this while also adding more information to key sections highlighted by the reviewers. We have moved some material from Sections **2** and **3** to the supplementary material, and removed five tables and two figures.

[RC 2] The other main point, maybe more fundamental, is the following one: I'm wondering what do we finally learn from this study? Although I appreciate and recognize the important efforts made to collect information from a large set of diverse sources and interacting with stakeholders, the result is a relatively simple matrix which I consider to be a bit thin for a journal paper. This point becomes especially acute if you consider that this same matrix and framework was already developed and presented in the previous Gill and Malamud 2016 and 2017 papers. Do the authors think it is justifiable to yet publish another paper which presents basically the same result with (in my opinion) only little additional substance by applying it to Guatemala? The substance may actually be there, i.e. in the many sources studied, but it is currently hardly in the paper. The authors may therefore reconsider how they present what they have researched (cf my comments below). For instance, I hoped to find more quantitative information regarding the physical processes, e.g. how often do such interactions occur? I'm aware that with the approach taken providing quantitative information related to the physical processes may not be so evident but I would like to encourage the authors to think about it.

> [AC2] We believe that this paper builds on the global approaches that we set out in our Gill and Malamud (2014, 2017) papers, and refines/applies these to help characterise potential hazard interactions at national and sub-national spatial scales. While the matrices take the same visual form as the ones in Gill and Malamud (2014), the approach we have used to construct and populate these

matrices are significantly enhanced. We would emphasise that what we are presenting in this paper is not just a matrix output, but also a discussion of a process to go from global to regional scales (interdisciplinary, multi-method approach) that enables the development of comprehensive, systematic and evidenced overviews of potential hazard interactions. We are presenting a suite of visualisations that build on our Gill and Malamud (2014 and 2017) papers with a greater range of hazard types, and matrices that are populated using different evidence. In Gill and Malamud (2014), done at a 'global' and high-level scale, we relied on published literature and case studies, whereas in our NHESS manuscript we have integrated diverse evidence types including interviews, data generating workshops, and interrogation of civil protection bulletins.

We therefore point to the substance of this paper being:

- Development and description of an interdisciplinary, multi-method approach that enables the development of comprehensive, systematic and evidenced overviews of potential hazard interactions at a regional (e.g., national/sub-national) scale. This contrasts with the existing studies of potential interactions, which are generally selective about which hazards they include and do not describe the evidence for including/excluding certain hazards or interactions between hazards.

- Application of this approach in the context of Guatemala to produce a suite of comprehensive and robust frameworks of potential hazard interactions for two spatial scales (national and sub-national), and describes their application to disaster risk reduction (including through initial efforts to embed them into key government agencies in Guatemala). The matrices presented include 21 to 33 hazards (compared to 6 to 11 natural hazards in the surveyed literature examples, which we summarize in **Table 1** of our manuscript).

- Description of an approach for contrasting current individual/collective knowledge with the published regional interaction framework, using Matthews' Correlation Coefficient. The results of this both underpin *why* developing comprehensive and evidenced frameworks of interactions is important (highlighting knowledge gaps), and provides a tool (and we recognize other tools could also be used) to monitor changes in understanding of hazard interactions over time.

We recognise that there are additional layers of information that could be helpfully added to what we have currently presented (e.g., how often each interaction occurs). We do not currently have this information in a level of detail that would be helpful to the reviewer or reader, and it was beyond the scope of our initial research remit (which we have set out with more clarity in **Section 1**). For each interaction (not hazard), understanding the frequency-magnitude of occurrence and the range of potential impacts would involve significant work and collation of diverse information that we currently do not have (as noted on page 21 of our original manuscript).

We have highlighted some of the above directions and intent of our work in our introduction, and enhanced the discussion in **Section 4** of quantitative characterisation of hazard interactions, noting current challenges in doing this and how future work could help to enrich this characterisation of potential interactions. We have suggested one approach is using an online wiki-style system where users can click on a cell in the matrix, and upload relevant papers, datasets, or their own assessments of frequency-magnitude to help improve this understanding. We have also emphasized what readers outside of the case-study area of Guatemala might learn from our study.

We note that in **Section 4.2** we have set out a quantitative characterisation of hazard and disaster professionals' individual and collective knowledge of hazard interactions – and outlined the significance of this assessment.

[RC3] Most of, but not all, the interactions are quite obvious and well known, such as storms generating floods and landslides. In fact, most of the paper, including the matrices, focus on two interacting processes but the most interesting aspect I found were the cascading hazards (more than 2 processes involved) but unfortunately they receive only little space. Is it possible to extend this issue, beyond the two case studies (and possibly at the expense of sections 2 and 3 which could be shortened)?

[AC3] (a) The reviewer notes that hazard interactions in **Figures 3** are obvious. We accept that many hazard pairings included in the matrix (e.g., storms triggering floods, earthquakes triggering landslides), and their spatial relevance to Guatemala, are well known. However, we note that this paper is establishing an evidenced framework (expressed as a matrix) of potential interactions. This national scale framework of potential interactions is, we believe, rarely discussed in the current hazard literature, as well as the method for developing a comprehensive and systematic framework. This manuscript has also contrasted the full list of potential interactions in **Figure 3**, with individual and collective knowledge (**Section 4.2**) in the region. The results of this highlight that the spatial relevance of the interactions are not always obvious, and therefore a systematic documentation and visualisation of potential interactions could help. We have made these points clearer in the text. (b) While the matrices focus on how any one single hazard could trigger or increase the probability of another single hazard, it is possible to use these matrices to extract examples of longer cascades. We have made this clearer in the text. We have expanded **Section 3.4** on networks of hazard interactions. We have included an additional extended example, and an expanded discussion of the importance of such networks.

[RC4] Finally, my impression was that some more reflection is needed by the authors. The paper sometimes has more project report character, leaving the reader with a feeling that the authors were short of time. One would like to see more synthesis and less details that are often not particularly relevant. I suggest that the authors take sufficient time to reflect on the objectives and the research questions (both not mentioned in the text) and what can be learned; also how this study contributes to scientific progress. Especially the last point is not evident for me and is not addressed in the paper either. Overall, I'm not sure whether the authors will be able to revise the paper in a round of major revisions in a way that is in my opinion needed, or whether they would rather like to take their time to re submit it at a later stage.

[AC4] We have addressed this comment by reviewing both the introduction and the discussion sections, and improving the way in which we frame the work we have done.

- We have clearly articulated our research questions (see the opening cover letter, pp. 1 to 2, and **AC2**) in the introduction, and in doing so help to frame the subsequent sections.
- We have expanded the discussion section to outline how this manuscript advances both multi-hazard research, and disaster risk reduction practice.

A key step in understanding risk (Sendai Framework Priority for Action 1) is understanding the hazard landscape of a region (i.e., the relevant single hazards, and how they may interact to generate combinations or cascades of interactions). Currently, regional studies of potential hazard interactions are sparse and none of these set out a replicable and scalable method for systematically doing this. In our paper, we describe and apply an approach that is replicable and can be applied at regional, national and sub-national spatial extents.

In the multi-step, method we present through an application to Guatemala, we include 2 to 3 times the number of natural hazards that other regional studies have included, and (rarely done) comprehensively set out the evidence for these interactions being spatially relevant. We integrate evidence from both natural and social science methodologies to construct a visualisation that – when returned to Guatemala – was shown to provoke cross-hazard and cross-institutional dialogue.

We believe this supports the scientific community to help construct more evidenced and detailed profiles of relevant interactions for diverse user groups, and through these profiles identify specific research and innovation gaps, as well as knowledge exchange and collaboration opportunities. We have integrated some of these comments, and expanded on them, in Section 4.

*Specific comments:*

[RC5] Introduction: I think this section could benefit from more text on the processes. The complexities of interacting hazard processes seem to find little attention.

[AC5] We have added further detail on the complexities of interacting hazard processes to the introduction.

[RC6] Section 3.3.1: this is an example of a section which is quite confusing to read. The six points made towards the end are not really clear and are they needed?

[AC6] We have moved some information into the Supplementary Material to improve the clarity of **Section 3.3.1**.

[RC7] Section 3.4: as mentioned, I found this the most interesting (and probably novel) section but it is not strongly developed. Is a more quantitative analysis possible?

[AC7] We have expanded **Section 3.4** on networks of hazard interactions. We have included an additional extended example from the evidence collected, and an expanded discussion of the importance of such networks. This results in (i) profiling of more examples from Guatemala, and (ii) synthesising key implications for disaster risk reduction from such examples. While we do not have the data to apply such a method to any of the scenarios we present, we can point the reader here to existing methods for quantitatively assessing probabilities of specific hazard cascades (e.g., using event scenario trees, such as done by Neri *et al.* (2008, 2013).

[RC8] Page 17, lines 1-3: another option could be to work with / engage researchers with appropriate level of Spanish language.

[AC8] We acknowledge this is one approach that would work, and included reference to this in the text.

[RC9] Page 17, lines 4-12: a very important point in my experiences working in such sociocultural environments. It applies in particular if risks are considered.

[AC9] We agree, and felt it was helpful to emphasise in the write up given the importance of the natural science community being more aware of such considerations.

[RC10] Page 17, lines 25-28: what are the implications of this points?

[AC10] We asked workshop participants to describe two different types of interaction: (1) one hazard triggering another hazard, and (2) one hazard increasing the probability of another hazard. If confusion between these two types of interaction existed, it is possible that participants may have inadvertently characterised an interaction as one type when they meant the other (i.e., a specific hazard pairing suggested to be a triggering relationship may actually be means to be communicated as an increased probability relationship). We do not believe the results expressed in **Figures 3** and **6** are unduly influenced by this, given the use of multiple evidence sources to construct them. We have changed some wording to clarify this.

[RC11] Discussion section: has interesting and important elements for people working in similar environments. As indicated above, I would like to see more reflection on how this paper advances research on multi-hazards.

[AC11] Our discussion section currently explores how regional interaction frameworks can advance multi-hazard risk reduction. We have focused more on the relationship of our work to practice, rather than research, but can expand the discussion to include more on the latter. We currently explore collective knowledge of hazard interactions in Guatemala, and note that interaction frameworks help to facilitate enhanced cross-institutional dialogue about hazard interactions, their likelihoods and potential impacts. This could help to strengthen collective knowledge of hazard interactions, and the ability of an individual to access this knowledge. We also described the response of hazard and civil protection professionals in Guatemala to our results, and their perspectives on 'next steps'. We also briefly describe in **Section 4** how interaction frameworks can help to improve decision making in key agencies engaged in DRR and civil protection. We have expanded this section to further outline how this work advances multi-hazards research. For example, interaction frameworks can guide future research priorities by determining where there is a lack of evidence and/or understanding of certain interactions. See also **AC2, AC4**, **AC36**.

[RC12] Figure 1: not sure this Figure is needed. Considered that the hazard codes need to be explained which is only done is subsequent figures.

[AC12] We have removed Figure 1, and integrated relevant information into old Figure 2.

[RC13] Figure 5: many place words are not particularly well readable.

[AC13] We have edited the figure to try and make the text larger and clearer.

[RC14] Figure 9: I was wondering whether the color code and the symbols are really used in this figure?

[AC14] We have adjusted the legend in this figure to remove this information.

[RC15] Table 8: I appreciate the level of detail in this table. But it was not clear to me how the hazard sub-types are then used? It is rather just a list which has a value in its own but no further relevance for the paper?

[AC15] The list presented in **Table 8** was developed from the evidence described in **Section 2**, as a classification of hazards relevant to Guatemala, using categories that many stakeholders in the region would understand. We take our classification and use this as the basis for the analysis in **Figures 6** to **8**. We have left **Table 8** in the manuscript, but added a note to **Section 3.2** to outline how this classification is integrated into the rest of the paper.

[RC16] Table 10: I'm not sure how well this table informs us. I found it rather confusing. We see the different bulletin reports which are not necessarily in a logical order (reflecting some issue there) and then the narrative summary. What is really the purpose of this table?

[AC16] The purpose of this table is to demonstrate an approach for identifying relevant, complex cascades that have previously impacted Guatemala. We highlight that while evidence exists for these cascades in a set of civil protection bulletins, they are not outlined in a coherent way but often different strands are included in different bulletins. **Table 10** presents four examples of the cascades that the reviewer highlighted to be particular interesting in **RC3**. We include the bulletin information to connect these examples to the evidence that describes them, but accept that the event description and narrative summary could be combined to make the table more succinct. In **AC3**, we note that we have expanded **Section 3.4** to include a more detailed discussion of cascades. We have also revised the text introducing **Table 10** to better articulate its purpose.

[RC17] Table 11: I'm not convinced that this table and information needs to be part of the paper (and then probably the respective section as well). Please re-consider.

[AC17] We believe that it is important to make reference to anthropogenic processes in this paper, given their ability to trigger and/or catalyse natural hazards. Many stakeholders emphasised the importance of anthropogenic activity in triggering landslides in Guatemala, for example. Reviewer 2 also noted this to be an interesting section (see **RC35**). We have kept the section short and signpost to other literature. We have moved **Table 11 t**o the supplementary material.

**Reviewer 2: Kirsten v. Elverfeldt**

[RC18] Summary. The paper deals with the development of regional interaction frameworks for Guatemala by utilizing literature reviews, field observations, interviews, and workshops. With the information thus gathered, a classification scheme of natural hazards is determined. Matrices were used to further determine hazard interactions, with a strong focus on the interaction (triggering or increasing the possibility) of two hazards.

Review summary

1. Does the paper address relevant scientific questions within the scope of NHESS? The paper falls into the subject areas of NHESS. It might fit the scope to understand the behaviour of hazardous natural events.

**2. Does the paper present novel concepts, ideas, tools, or data? This remains rather unclear since the authors do not explicitly state the aims, research questions, hypotheses, and novelties of the paper.**

**3. Are the scientific methods and assumptions valid and clearly outlined? Assumptions are not made explicit. Methods are valid and transparently explained, but explanations would need streamlining and re-structuring.**

**4. Are the results sufficient to support the interpretations and conclusions? Yes, though the novelty of the results needs to be stressed. I have the feeling that there could be more to the paper than the authors actually delivered. It is difficult to review this paper because the authors leave it to the reader to "read between the lines" and to draw conclusions by herself/himself. In a nutshell, it remains somewhat unclear what we gain by the paper.**

5. Do the authors give proper credit to related work and clearly indicate their own new/original contribution? Yes.

6. Does the title clearly reflect the contents of the paper? Yes.

**7. Does the abstract provide a concise and complete summary? Yes. However, in the abstract research questions, hypotheses, aims,: : : are missing (as they are in the text).**

**8. Is the overall presentation well-structured and clear? No. Needs to be improved.**

9. Is the language fluent and precise? Yes.

10. Are the number and quality of references appropriate? Yes.

[AC18, addressing the comments highlighted in bold above]. The reviewer notes that the manuscript needs some streamlining and restructuring, clearer articulation of assumptions, and added emphasis on the novelty and importance of the results. We have:
- Revised the abstract and introduction (**Section 1**), including making our objectives and research questions clearer, as articulated in the cover letter of this response.

- Revised the introduction (**Section 1**) and discussion sections (**Section 4**) to make it clearer what the novelty of this paper is, and how this advances disaster risk reduction in multi-hazard contexts (see **AC2** and **AC4**).

[RC19] Page 2, Line 3: The authors start very abruptly with the topic of regional interaction frameworks, without really framing their topic. They present the term "regional interaction framework" right at the beginning, whilst the definition of the term only comes one paragraph later.

[AC19] We have expanded the introduction, with more framing of the topic, and its relevance to multi-hazard approaches. We have included a definition of the term *regional interaction framework* immediately after its first use.

[RC20] Page 2, Line 4: It remains unclear in how far your approach is interdisciplinary. Even more so, it remains unclear what "the approach" is that is being applied. I suggest that at least (!) a citation of the previous Gill and Malamud papers on this subject should be given here; it'd be even better to continue (after framing your topic) with briefly explaining what your approach is. In general, the writing style of section 1.1. is rather additive than providing an argument for why the study is relevant or in what context it is to be understood. The aim of the paper remains unclear as well as hypotheses, assumptions, and research questions.

[AC20] In our rewriting of the introduction (see **AC5**, **AC18**, **AC19**), we have set out the approach that we are following and better articulated our research questions.

[RC21] Page 3, Line 13: Is Table 2 necessary? Please consider deleting the table.

[AC21] We have moved this table to the supplementary material.

[RC22] Page 3, Line 20: Here, you distinguish between hazard interactions on the one hand and networks of interactions (cascades) on the other hand, whilst on page 2 you summarized all interrelated effects (including cascades) under the umbrella of the term hazard interactions. Please consider handling this coherently. To me, section 1 is rather overstructured. For example, section 1.3 consists of only three sentences. I'd suggest to re- and de-structure the section, including a better framing of the topic and to be less descriptive and additive, and to put up an argumentation.

[AC22] We have reviewed language and tried to be consistent throughout the. As noted in **AC5** and **AC19** we have expanded the introduction and included more material to frame this discussion. This has resulted in **Section 1** being restructured. Our writing style preference is to retain a 'structure of paper' section at the end of the introduction to guide the reader.

[RC23] Page 4, Line 12: Suggestion to delete Table 4.

[AC23] We have deleted this table, and included the content within the manuscript text.

[RC24] Page 5, Line 30f: I'd also suggest deleting Table 5.

[AC24] We have deleted this table, and included the content within the manuscript text.

[RC25] Page 7, Line 6: Suggestion to delete Table 6

[AC25] We have deleted this table, and included the content within the manuscript text.

[RC26] Page 8, Line 16: Here, and at quite a few instances before and afterwards, you refer to later sections in the paper. This makes reading rather difficult and raises the question whether the paper could be structured

more coherently. If you discussed the workshops in section 2.6, why do you discuss their limitations so much later in the paper? As a rule of thumb references to content delivered later in a paper should be avoided.

[RC27] Page 9-10, Lines 15ff: In the paper, comparatively long sections are dedicated to referencing to previous or later content. Suggestion to shorten and re-structure the paper.

[RC28] Page 10, Line 8ff: this explanation of what is required for regional interaction frameworks comes at a rather late stage. Since you mention regional interaction frameworks so often on previous pages, I'd suggest to bring together issues that belong together. This would also decrease the amount of references to previous and later sections in the paper. The paper in its current stage is rather difficult to read and readers might easily lose track of what is the intention of the paper or a section in its own.

[AC26] [AC27] [AC28] We have reviewed references to previous and future content and tried to reduce this. We think some of this referencing can be helpful, to signpost to the reader that we are building on something that has come previously. We have grouped detail on regional interaction frameworks in Section 2.8

[RC29] Page 10, Line 19: this has been mentioned before (on page 2)

[AC29] We have rephrased **Section 3.1** so it better builds on what was presented earlier in the manuscript.

[RC30] Page 12, Line 4: In table 8, A-E are named differently from what was proposed in the text.

[AC30] This is now corrected.

[RC31] Page 12, Line 15: Figure 4: I am not sure that it is useful to have the same figure as in figure 3 repeated only to deliver the information of how many evidence sources were used. I think it is enough to deliver this information via text only (the number of figures and tables is really high for this paper, and not all of them seem to be necessary).

[AC31] The purpose of **Figure 4** is to rapidly assess where there could be uncertainty, and future research needed. We do not think this would be easy if the information was presented in the text. We tried to add additional information to **Figure 3**, but this reduces the clarity of this key figure. We have moved **Figure 4** to the Supplementary Material, and referred to it in the figure caption of old **Figure 3**.

[RC32] Page 13, Line 14: Figure 5 – again, I'd expect to get this information much earlier, e.g. in section 1.2. In table 9, evidence categories A-E differ again from text

[AC32] We have moved **Figure 5** to the introduction. A-E are now consistent between **Table 9** and the text.

[RC33] Page 13, Line 24: Figure 6 text is too small, rather impossible to read; is it upside down?

[AC33] We have increased the text size and reduced the amount of information presented in this figure to increase its clarity.

[RC34] Page 14, Line 18ff: I cannot quite see the difference between example 1 and 4 (Table 10)? It would also be helpful if you explained what you mean by "linear event", "multi-branch event" etc. This again is some example for how you (superficially) describe rather than explain or argue.

[AC34] We have added further explanation to what we mean by these terms, and enhance the explanations in this section as noted in previous comments (**AC3** and **AC16**). Examples 1 and 4 do

have some similarities, and we have removed one from **Table 10** to help make the discussion more succinct. We will also include a simple, visual summary of each example to illustrate the example.

[RC35] Page 15, Line 13: In table 11, evidences A-E differ from text

[AC35] This is now corrected.

[RC36] Page 16, Line 1: It would be useful if you explained and/or detailed the "useful insights" that are generated. I really do like the way you collate information via different methods (literature, interviews, workshops etc.). But I think your paper stops when it gets most interesting, i.e. hazard cascades/networks and anthropogenic impacts on hazard interactions. Furthermore, since you do not explain what you gain aside from a visualisation and collection of (maybe more or less) known hazard interactions, this important aspect remains far too vague. This might also be because the reader doesn't know your aims, hypotheses, and research questions.

[AC36] We refer to our reply to **AC2** and **AC4**. We would emphasise that what we have gained extends beyond the location-specific visualisation, to a replicable and scalable method that can be applied in other contexts to better understand the hazard landscape. A comprehensive overview of potential hazard interactions allows agencies responsible for hazard monitoring and response to assess if current disaster risk reduction and response strategies, and communication and collaboration mechanisms, can be enhanced to recognise the complexity represented in this paper. We have extended **Section 4** to better articulate the significance of what we have developed and how this can be used (and augmented) to improve disaster risk reduction, along with signposting the relevance of our work in the introduction and other selected places in our manuscript. For example, we have added to **Section 4.4** to describe how interaction frameworks can help to improve decision making in key agencies engaged in DRR and civil protection, such as guiding future research priorities by determining where there is a lack of evidence and/or understanding of certain interactions.

[RC37] Page 16, Line 10: This is another example that re-structuring the paper is necessary. The limitations and uncertainties should be mentioned where you present the respective method; here, you can then focus on the discussion.

[AC37] Many of the limitations we present cut across multiple evidence types, and therefore the limitations are more succinctly described when presented together. We have, however, moved these to end of **Section 2** so that they naturally come after the descriptions of evidence types.

[RC38] Page 16, Line 29: I'm confused by the additional information about translators – have you used them? If not, why? If you did, this should be mentioned earlier.

[AC38] We used a variety of translation methods, and have added a line about this in **Section 2.5** (Stakeholder Engagement: Interviews)

[RC39] Page 17, Line 25ff: Plus, if you use a pre-defined hazard scheme without the option to add other hazards and interactions, participants' knowledge might be missed out.

[AC39] A line was added to the list of limitations.

[RC40] Page 18, Line 22: Table 9 – colour code and symbol code (legend) to be deleted

[AC40] We have removed this information from **Figure 9**.

[RC41] Page 19, Line 17: Why did you set these thresholds and not others? Explanation would be good.

[AC41] Thresholds of 3 and 5 were selected arbitrarily to demonstrate how this approach could be adjusted to remove those interactions only volunteered by one (or a small number of) professionals, thus acting as a form of quality control. We could have chosen thresholds of 2 or 4, but determined that increments of 1, 3 and 5 would give a spread of results to illustrate the discussion. We do not place great emphasis on the specific threshold in the manuscript, nor try to defend this as being a critical choice. Rather we demonstrate how this approach can help to examine differences between stakeholder perspectives and our national interaction frameworks, and monitor changing understanding and perceptions of natural hazard interactions. We have added a note regarding this to the text.

[RC42] Page 1, Line 11: , and evidenced : : :

[RC43] Page 1, Line 15f: to reduce the number of parentheses, I'd suggest to re-write a part of the sentences as follows: (internationally accessible: 93 peer-review and 76 grey literature sources); (locally accessible civil protection bulletins: 267 bulletins from 11 June 2010 to 15 October 2010)

[RC44] Page 2, Line 3: , and evidenced : : :

[RC45] Page 2, Line 13: Delete "Here, and"

[RC46] Page 2, Line 27: Put "and" in italics (two times)

[RC47] Page 2, Line 29: Consider rephrasing: : : :"that our approach also supports implementation"

[RC48] Page 3, Line 5: , and surface collapses

[RC49] Page 3, Line 7: , and cold spells

[RC50] Page 3, Line 23: for coherence, I'd suggest to change the heading to ": : : regional interaction framework"

[RC51] Page 3, Line 25: evidences [current wording is ok]

[RC52] Page 4, Line 3: , and media reports (please check for "comma + and" throughout the document).

[RC53] Page 4, Line 6: consider rephrasing "an overview of Guatemala's hazard-forming"

[RC54] Page 4, Line 15: include is repetitive in the sentence

[RC55] Page 4, Line 17: verb missing?

[RC56] Page 7, Line 2: helped identifying [Corrected to 'helped to identify']

[RC57] Page 7, Line 7: selected locations [current wording is ok]

[RC58] Page 16, Line 9: Delete "."

[RC59] Page 19, Line 2: "." is missing

[AC42–59] We have made these corrections, or confirmed that the current wording is correct (RC51 and RC57).

[revised manuscript text omitted]
 can help to inform urban planning by complementingcreating scenarios where the overlay of multiple single-hazard maps to explore appropriate land-use. Recognising the there are potential for interactions between spatially overlapping or contiguous hazards. This can then help to ensure risk is not underestimated and build effective hazard management plans that take into account potential cascades of hazards. For example, an underground transport system may need to consider how an earthquake triggering subsidence would affect its susceptibility to groundwater flooding.

*Scenarios to eEnsureing disasterhazard preparedness and disaster response systems are effective.* The regional interaction framework can be a powerful tool for scenario discussions between hazard managers and those responsible for single-hazard preparedness and response, particularly in the context of Characterising multi-hazard landscapes and interaction networks can help to inform disaster response planning. 
[revised manuscript text omitted]

| Example | Bulletin # | Bulletin Date | Location | Event Description Narrative Summary | Visual Summary |
|---|---|---|---|---|---|
| | 1199 | | Centre and South Guatemala |  A warning was issued that Storm Matthew could trigger damage, and was associated with flash floods, landslides and mudslides in Nicaragua and Honduras. On 25 September 2015, Tropical Storm Matthew  impacted Guatemala directly, causing river levels to rise and saturate soils, with a warning that flooding may occur. The next bulletins reported flooding, an increased likelihood of landslides, and lightning. | |

**Table 11. Relevant anthropogenic process types in Guatemala.** A description of the four evidence types A–E, together with additional references, used to identify 17 anthropogenic process types as being spatially relevant in Guatemala.

| Anthropogenic Process Type | Evidence A = International Literature B = Civil Protection Bulletins C = Field Observations D = Stakeholder Interviews E = Workshop (*anthropogenic processes not discussed*) * (Reference) = Additional citations, beyond A–E. | | | |
|---|---|---|---|---|
| Groundwater Abstraction | | | D | |
| Oil/Gas Extraction | | | | * (OEC, 2016) |
| Subsurface Infrastructure Construction | A | | D | |
| Subsurface Mining | | | | * (OEC, 2016) |
| Material (Fluid) Injection | | | | * (USGeothermal, 2016) |
| Vegetation Removal | A | C | D | |
| Agricultural Practice Change | | C | D | |
| Urbanisation | | C | D | |
| Infrastructure Construction (Unloading) | | | D | |
| Quarrying/Surface Mining (Unloading) | | | | * (OEC, 2016) |

| Anthropogenic Process Type | Evidence
A = International Literature
B = Civil Protection Bulletins
C = Field Observations
D = Stakeholder Interviews
E = Workshop (*anthropogenic processes not discussed*)
* (Reference) = Additional citations, beyond A–E. | | | |
|---|---|---|---|---|
| Infrastructure (Loading) | | | C | D |
| Infilled (Made) Ground | A | | | |
| Reservoir and Dam Construction | A | | D | * (Salini Impregilo, 2014) |
| Drainage and Dewatering | A | B | D | |
| Water Addition | A | B | D | |
| Chemical Explosion | | | | *Inferred relevant* |
| Fire | | | D | |

**Table 7. Calculation of Matthews' Correlation Coefficient (*MCC*) to assess agreement between the collective knowledge of 16 workshop participants (Figure 2) and national interaction framework (Figure 5).** Three different thresholds, each relating to the number of workshop participants (out of 16) identifying a particular interaction, are used to determine collective knowledge of hazard interactions. The number of 'agreements' and 'disagreements' between the workshop participants' response and national interaction framework (see column headers for descriptions) is shown. For each row, the sum of True Positives (TP) and False Negatives (FN) is 50, and the sum of True Negatives (TN) and False Positives (FP) is 392. *MCC* values are determined using **Eq 1**. An *MCC*  +1.0 means complete agreement; an *MCC*  −1.0 means complete disagreement.

| Workshop Participants Identifying an Interaction *(n = 16)* | # Interactions Identified by ≥ x participants (TP + FP) | AGREEMENT [Participants' Collective Framework and National Interaction Framework Agree] | | DISAGREEMENT [Participants' Collective Framework and National Interaction Framework Do Not Agree] | | Matthews' Correlation Coefficient (Eq 1) |
|---|---|---|---|---|---|---|
| | | Interaction Occurs in Both Frameworks | Interaction Does Not Occur in Either Framework | Interaction Occurs in National Framework but not Participants' Collective Framework | Interaction Occurs in Participants' Collective Framework but not National Framework | |
| | | True Positives (TP) | True Negatives (TN) | False Negatives (FN) | False Positives (FP) | |
| ≥ 1 | 86 | 25 | 330 | 25 | 61 | 0.28 |
| ≥ 3 | 32 | 22 | 381 | 28 | 10 | 0.51 |
| ≥ 5 | 19 | 16 | 388 | 34 | 3 | 0.49 |

---

## Author Response (AR2)

**British Geological Survey**
Expert | Impartial | Innovative

**Keyworth**

Environmental Science Centre
Keyworth
Nottingham
United Kingdom
NG12 5GG

Telephone   +44(0)115 9363100
Direct Line  +44(0)115 9363138
E-mail        joell@bgs.ac.uk
Web           www.bgs.ac.uk

11 November 2019

**Re: nhess-2018-363**

Dear Professor Fuchs,

Please find enclosed the *nhess-2018-363* manuscript for production of the final article by Copernicus. Please note that the version uploaded has minor differences to the article accepted for publication, largely in terms of final changes to adhere to your publishing guide. We have not changed the substance or argument of the paper in any minor or major way.

The primary change we have made is to the title of the paper. We have changed:

*Regional Interaction Frameworks to Support Multi-Hazard Approaches to Disaster Risk Reduction (With an Application to Guatemala)*

To a shorter and more succinct title:

*Construction of regional multi-hazard interaction frameworks, with an application to Guatemala.*

Minor formatting changes include: (i) replacing *et al.* with et al. (ii) replacing Figure 1 with Fig. 1 and Section 1 to Sect. 1, (iii) reducing the use of italics, (iv) changing 'hazard' to 'natural hazard' where we think it makes the meaning clearer to the reader, (v) other small changes to sentence structure to improve the clarity (but not meaning) of the text.

The changes made are largely to support the copy-editing process, and ensure all requested sections (e.g., author's contributions) are included. Please do not hesitate to contact me if you have any questions.

Yours sincerely,

Joel C. Gill PhD FGS AFHEA
*International Development Geoscientist, British Geological Survey*

[Figure]

[Figure]
 disability confident EMPLOYER

INVESTOR IN PEOPLE